# An Improved Analysis of Alternating Minimization for Structured Multi-Response Regression

**Sheng Chen** [*]
The Voleon Group
chen2832@umn.edu

**Arindam Banerjee**
Dept. of Computer Science & Engineering
University of Minnesota, Twin Cities
banerjee@cs.umn.edu

## Abstract

Multi-response linear models aggregate a set of vanilla linear models by assuming correlated noise across them, which has an unknown covariance structure. To find the coefficient vector, estimators with a joint approximation of the noise covariance are often preferred than the simple linear regression in view of their superior empirical performance, which can be generally solved by alternating-minimization-type procedures. Due to the non-convex nature of such joint estimators, the theoretical justification of their efficiency is typically challenging. The existing analyses fail to fully explain the empirical observations due to the assumption of resampling on the alternating procedures, which requires access to fresh samples in each iteration. In this work, we present a resampling-free analysis for the alternating minimization algorithm applied to the multi-response regression. In particular, we focus on the high-dimensional setting of multi-response linear models with structured coefficient parameter, and the statistical error of the parameter can be expressed by the complexity measure, Gaussian width, which is related to the assumed structure. More importantly, to the best of our knowledge, our result reveals for the first time that the alternating minimization with random initialization can achieve the same performance as the well-initialized one when solving this multi-response regression problem. Experimental results support our theoretical developments.

## 1 Introduction

We consider the following multi-response linear model [1, 5, 18] with $m$ real-valued outputs,

$$\mathbf{y} = \mathbf{X}\boldsymbol{\theta}_* + \boldsymbol{\eta}, \quad \text{where } \boldsymbol{\eta} = \boldsymbol{\Sigma}_*^{1/2}\tilde{\boldsymbol{\eta}} \tag{1}$$

where $\mathbf{y} \in \mathbb{R}^m$ is the response vector and $\mathbf{X} \in \mathbb{R}^{m \times p}$ consists of $m$ $p$-dimensional feature vectors, and $\tilde{\boldsymbol{\eta}} \in \mathbb{R}^m$ is a zero-mean isotropic noise vector. The $m$ responses share the same underlying parameter $\boldsymbol{\theta}_* \in \mathbb{R}^p$, which corresponds to the so-called *pooled model* [17]. Without loss of generality, the counterpart of (1) with response-specific parameters can be equivalently written in the above form, by block-diagonalizing rows of $\mathbf{X}$ and concatenating different parameters into a single vector. What makes this model different from vanilla linear models is the correlated noise $\boldsymbol{\eta}$ across responses, which is assumed to be a linear transformation of $\tilde{\boldsymbol{\eta}}$. The noise covariance of $\boldsymbol{\eta}$ is given by $\text{Cov}(\boldsymbol{\eta}) = \boldsymbol{\Sigma}_*$. This model has found numbers of real-world applications, such as econometrics [17], computational biology [22] and climate informatics [14, 15], just to name a few.

In practice, we are given $n$ observations of $(\mathbf{X}, \mathbf{y})$, denoted by $\mathcal{D} = \{(\mathbf{X}_i, \mathbf{y}_i)\}_{i=1}^n$, while the noise covariance structure $\boldsymbol{\Sigma}_*$ between responses is typically unknown. Our goal is to estimate the

---

[*]This work was done when the author studied at University of Minnesota, Twin Cities

parameter $\boldsymbol{\theta}_*$, ideally together with $\boldsymbol{\Sigma}_*$. In this work, we additionally focus on the high-dimensional regime, where the true parameter $\boldsymbol{\theta}_*$ is assumed to possess certain low-complexity structure measured by some function $f : \mathbb{R}^p \mapsto \mathbb{R}_+$, which can be either convex (e.g., norms) or non-convex (e.g., $L_0$ cardinality). For the low-dimensional setting, it has been shown, both empirically and theoretically, that simultaneously estimating $\boldsymbol{\Sigma}$ and $\boldsymbol{\theta}$ leads to better performance than ordinary least squares [20]. Inspired by this fact, we consider the joint estimator of $(\boldsymbol{\Sigma}, \boldsymbol{\theta})$ in high dimension as follows,

$$\left(\hat{\boldsymbol{\theta}}_n,\, \hat{\boldsymbol{\Sigma}}_n\right) = \underset{\boldsymbol{\theta} \in \mathbb{R}^p,\, \boldsymbol{\Sigma} \succeq 0}{\operatorname{argmin}} \frac{1}{2n} \sum_{i=1}^{n} \left\| \boldsymbol{\Sigma}^{-\frac{1}{2}} \left(\mathbf{y}_i - \mathbf{X}_i \boldsymbol{\theta}\right) \right\|_2^2 + \frac{1}{2} \log |\boldsymbol{\Sigma}| \quad \text{s.t.} \quad f(\boldsymbol{\theta}) \leq \lambda\,, \quad (2)$$

which corresponds to the constrained maximum likelihood estimator (MLE) of $(\boldsymbol{\Sigma}, \boldsymbol{\theta})$ when the noise is multivariate Gaussian. The structural assumption on $\boldsymbol{\theta}_*$ is encoded by the inequality constraint. Though the noise structure is accounted in this joint estimator, one challenge faced by the associated optimization problem is the non-convexity of the objective function. In the light of the simplicity of the marginal optimization over $\boldsymbol{\Sigma}$ and $\boldsymbol{\theta}$ when the other is fixed, a popular approach to dealing with such problem is *alternating minimization* (AltMin), i.e., alternately solving for $\boldsymbol{\Sigma}$ (and $\boldsymbol{\theta}$) while keeping $\boldsymbol{\theta}$ (and $\boldsymbol{\Sigma}$) fixed. For problem (2), the update of AltMin can be written as

$$\hat{\boldsymbol{\Sigma}}_{(t+1)} = \frac{1}{n} \sum_{i=1}^{n} \left(\mathbf{y}_i - \mathbf{X}_i \hat{\boldsymbol{\theta}}_{(t)}\right) \left(\mathbf{y}_i - \mathbf{X}_i \hat{\boldsymbol{\theta}}_{(t)}\right)^T\,, \quad (3)$$

$$\hat{\boldsymbol{\theta}}_{(t+1)} = \underset{\boldsymbol{\theta} \in \mathbb{R}^p}{\operatorname{argmin}} \frac{1}{2n} \sum_{i=1}^{n} \left\| \hat{\boldsymbol{\Sigma}}_{(t+1)}^{-\frac{1}{2}} \left(\mathbf{y}_i - \mathbf{X}_i \boldsymbol{\theta}\right) \right\|_2^2 \quad \text{s.t.} \quad f(\boldsymbol{\theta}) \leq \lambda\,, \quad (4)$$

which will be executed for a number of iterations, say $T$. Here the new $\hat{\boldsymbol{\Sigma}}_{(t+1)}$ is obtained by computing the empirical covariance of the residues estimated at $\hat{\boldsymbol{\theta}}_{(t)}$. Though $f$ can potentially be non-convex, the update of $\hat{\boldsymbol{\theta}}_{(t+1)}$ is merely solving a constrained least squares problem, for which various algorithms are guaranteed to find the global minimum under mild conditions on data [21, 3, 33]. Generally speaking, both steps are easy to implement, which makes AltMin more attractive compared with other optimization algorithms that jointly update $\boldsymbol{\theta}$ and $\boldsymbol{\Sigma}$. In the low-dimensional setting, the AltMin algorithm for multi-response regression was initially proposed by [28]. For the high-dimensional counterpart with sparse parameters, previous works [32, 23, 31] considered the regularized MLE approaches, which are also solved by AltMin-type algorithms. Unfortunately, *none* of those works provide *finite-sample* statistical guarantees for their algorithms. The first attempt to establish the non-asymptotic error bound of this AltMin approach is made by [20] for low-dimensional regime, with a brief extension to sparse parameter setting using iterative hard thresholding method [21]. But they did not allow more general structure of the parameter. One of the closely related work is [10] with a focus on general parameter structures captured by norms. They proposed an alternating estimation framework, in which the generalized Dantzig selector [8] is used for $\boldsymbol{\theta}$-step as an alternative to the regularized and the constrained estimators.

The AltMin technique has also been applied to many other estimation problems, such as matrix completion [19], phase retrieval [27], and mixed linear regression [44]. However, the current theoretical understanding of AltMin is still incomplete. Including the aforementioned works, the statistical guarantees for non-convex AltMin procedures are often shown under the *resampling* assumption, which assumes that each iteration receives a fresh sample. Albeit this can be achieved by partitioning the data into disjoint subsets and using different batches in each update, people seldom do so in practice, as it usually results in worse performance than using all data in every iteration. From the theoretical perspective, the resampling assumption oversimplifies the analysis of the algorithm used in practice, which may otherwise require sophisticated proof techniques [36].

In this paper, we aim at a better way to bound the statistical error of the above AltMin procedure for general structure-inducing $f$. In principal, non-asymptotic statistical analyses for the high dimension typically involve bounding suprema of stochastic processes [26, 2, 42, 30]. The difficulty of analyzing AltMin lies in the dependency between the data and the obtained iterates, and the lack of independence prevents applications of various concentration inequalities to the suprema of the target processes. The resampling assumption facilitates the analysis of AltMin by assuming access to new data that are independent of previous iterates. In contrast to resampling, we here resort to uniform bounds to tackle the dependency issue. That is, instead of dealing with the processes involving the specific iterates generated by AltMin, we try to bound their worst-case counterparts that consider *all* possible iterates

before running AltMin. This solution to dependency ends up with more complicated stochastic processes, which need careful treatment. By applying *generic chaining* [37], an advanced tool from probability theory, we are able to obtain the desired bounds for the processes under consideration, and eventually express the error bound in terms of a complexity measure called *Gaussian width* [16, 7] (see Section 3.1). In particular, we analyze the AltMin procedure under two different choices of initialization, one with an arbitrarily initialized iterate and the other starting at a point close to $\boldsymbol{\theta}_*$. The $L_2$-error for both types of AltMin is shown to converge *geometrically* to certain *minimum achievable error* $e_{\min}$ with overwhelming probability, i.e.,

$$\left\|\hat{\boldsymbol{\theta}}_{(T)} - \boldsymbol{\theta}_*\right\|_2 \le e_{\min} + \rho_n^T \cdot \left(\left\|\hat{\boldsymbol{\theta}}_{(0)} - \boldsymbol{\theta}_*\right\|_2 - e_{\min}\right) \tag{5}$$

where $\rho_n < 1$ is the contraction factor and $e_{\min}$ is given by

$$e_{\min} = O\left(\frac{w(\mathcal{C}) + m}{\sqrt{n}}\right) \quad \text{(arbitrary initialization)}, \tag{6}$$

$$e_{\min} = O\left(\frac{w(\mathcal{C})}{\sqrt{n}}\right) \quad \text{(good initialization)}. \tag{7}$$

Here $w(\mathcal{C})$ is the Gaussian width of a set $\mathcal{C}$ related the structure of $\boldsymbol{\theta}_*$ (see Definition 2). Surprisingly the error for good initializations matches the resampling-based result up to some constant, which requires more fresh data to achieve such a bound. In general, our work improves the results in [20, 10] in several aspects. First, our analysis does not rely on the resampling assumption. Second, our statistical guarantees work for general sub-Gaussian noise while [20] and [10] only considered Gaussian noise. Third, we allow the complexity function $f$ to be non-convex, whereas [10] required $f$ being a norm. Last but not least, our result suggests that when the amount of data is adequate for the error bound (6) to meet the requirement of good initialization, the AltMin with arbitrary initialization can achieve the same level of error as the well-initialized one. Although this type of guarantee for arbitrary initializations was discovered for other problems [43], it has not been revealed for the multi-response regression, and our proof technique is also different from the existing ones.

The rest of the paper is organized as follows. In Section 2, we outline our strategy for combating non-convexity and present the algorithmic details of the AltMin procedure for structured multi-response regression. In Section 3, we present the statistical guarantees for the AltMin algorithm under suitable probabilistic assumptions. We provide some experimental results in Section 4, and conclude in Section 5. All proofs are deferred to the supplementary material.

## 2    Strategy to Conquer Non-Convexity

For many statistical estimation problems, we can construct the estimator of the underlying model parameter $\mathbf{w}_*$, by minimizing certain loss function on the given sample $\mathcal{D}$,

$$\hat{\mathbf{w}} = \underset{\mathbf{w} \in \mathcal{W}}{\operatorname{argmin}} \, L(\mathbf{w}; \mathcal{D}). \tag{8}$$

In order to show the recovery guarantee for non-convex estimation, there are mainly two commonly-used strategies. One strategy is to show certain local convergence in a neighborhood $\mathcal{N}$ of the global minimizer $\hat{\mathbf{w}}$ of (8) [6, 29, 39, 45, 25]. With a proper initialization inside $\mathcal{N}$, subsequent iterates produced by some local search might be able to converge to $\hat{\mathbf{w}}$, whose statistical error is expected to be small. This strategy is particularly suitable for the noiseless setting, as $\hat{\mathbf{w}}$ is equal to $\mathbf{w}_*$, and most of the existing works use gradient descent type or its variants as workhorse algorithms. The other strategy is to show that there is no spurious local minima of $L$ under the assumed statistical models, so that any optimization algorithms that provably converge to local minima will suffice for a good estimation [34, 35, 4, 13, 24, 12].

For our multi-response regression problem, however, it is difficult to apply the aforementioned strategies. First, bounding the statistical error of the global minimizer is nontrivial in the noisy setting, especially when the objective $L(\mathbf{w})$ involves more than one set of variables like the multi-response regression, let alone characterizing the equivalence of all local minima. Second, the gradient-based local search is inefficient for the problem (2), since the update of $\boldsymbol{\Sigma}$ involves matrix inversion and projection onto positive semidefinite (PSD) cone. In contrast, AltMin procedure has a closed-form solution to $\boldsymbol{\Sigma}$-step, which is preferred in this setting.

In this work, we consider another strategy for the non-convex estimation in which $\mathbf{w}$ ($\mathbf{w}_*$) is composed of two parameters, $\mathbf{a}$ and $\mathbf{b}$ ($\mathbf{a}_*$ and $\mathbf{b}_*$). The loss $L$ is assumed to jointly non-convex over $\mathbf{a}$ and $\mathbf{b}$, but might be marginally convex w.r.t. $\mathbf{a}$ ($\mathbf{b}$) when $\mathbf{b}$ ($\mathbf{a}$) is fixed. When the marginal subproblems are easy to solve, alternating minimization procedure is appealing for the purpose of estimation, which is true for the multi-response regression. The AltMin algorithm executes the following updates,

$$\hat{\mathbf{a}}_{(t+1)} = \underset{\mathbf{a}\in\mathcal{A}}{\operatorname{argmin}}\ L(\mathbf{a}, \hat{\mathbf{b}}_{(t)}; \mathcal{D})\ , \qquad \hat{\mathbf{b}}_{(t+1)} = \underset{\mathbf{b}\in\mathcal{B}}{\operatorname{argmin}}\ L(\hat{\mathbf{a}}_{(t+1)}, \mathbf{b}; \mathcal{D})\ . \qquad (9)$$

The basic idea for showing the statistical guarantees of AltMin is to derive the statistical error bounds for both the $\mathbf{a}$- and $\mathbf{b}$-steps when the other parameter is fixed to the latest estimate. Since both subproblems in (9) are usually simpler, the separate errors might be easier to characterize than considered jointly, which are ideally of the form,

$$d_1\left(\hat{\mathbf{a}}_{(t+1)},\ \mathbf{a}_*\right)\ \leq\ e_1\left(d_2\left(\hat{\mathbf{b}}_{(t)}, \mathbf{b}_*\right)\right)\ , \qquad d_2\left(\hat{\mathbf{b}}_{(t+1)},\ \mathbf{b}_*\right)\ \leq\ e_2\left(d_1\left(\hat{\mathbf{a}}_{(t+1)}, \mathbf{a}_*\right)\right)\ . \qquad (10)$$

The function $d_1$ (respectively $d_2$) characterizes the closeness between $\hat{\mathbf{a}}_{(t+1)}$ and $\mathbf{a}_*$ ($\hat{\mathbf{b}}_{(t+1)}$ and $\mathbf{b}_*$), which is nonnegative with $d_1(\mathbf{a}_*, \mathbf{a}_*) = 0$ ($d_2(\mathbf{b}_*, \mathbf{b}_*) = 0$) but not necessarily a metric. The choice of $d_1$ and $d_2$ depends on the goal of analysis for the problem under consideration, and a suitable combination of $d_1$ and $d_2$ may facilitate the proof. The upper bound $e_1$ (respectively $e_2$) may depend on other quantities such as $n$, but our emphasis is the dependence on the estimation accuracy of $\mathbf{b}$ ($\mathbf{a}$). It is natural to expect that $e_1$ ($e_2$) will shrink as $\hat{\mathbf{b}}_{(t)}$ ($\hat{\mathbf{a}}_{(t)}$) moves closer to $\mathbf{b}_*$ ($\mathbf{a}_*$). Under this condition, we can apply the bounds in (10) alternatingly and recursively

$$d_1(\hat{\mathbf{a}}_{(T)}, \mathbf{a}_*) \leq e_1\left(d_2\left(\hat{\mathbf{b}}_{(T-1)}, \mathbf{b}_*\right)\right) \leq \dots\dots \leq \underbrace{e_1\left(e_2\left(\dots e_1\left(d_2\left(\hat{\mathbf{b}}_{(0)}, \mathbf{b}_*\right)\right)\dots\right)\right)}_{\text{composition of } T\ e_1(\cdot) \text{ and } T-1\ e_2(\cdot)} \qquad (11)$$

$$d_2(\hat{\mathbf{b}}_{(T)}, \mathbf{b}_*) \leq e_2\left(d_1\left(\hat{\mathbf{a}}_{(T)}, \mathbf{a}_*\right)\right) \leq \dots\dots \leq \underbrace{e_2\left(e_1\left(\dots e_1\left(d_2\left(\hat{\mathbf{b}}_{(0)}, \mathbf{b}_*\right)\right)\dots\right)\right)}_{\text{composition of } T\ e_2(\cdot) \text{ and } T\ e_1(\cdot)} \qquad (12)$$

which may imply the error of $\hat{\mathbf{a}}_{(T)}$ and $\hat{\mathbf{b}}_{(T)}$ under other metrics of interest as well. Compared with the previous strategies, one notable difference of our treatment is that we do not care about the *optimization* convergence of AltMin, as we neither characterize the error of any local minimizers of $L(\cdot)$ nor show any iterate convergence to those minimizers. Instead the ingredients we need are simply the *statistical* error bounds in (10). Given this fact, our analysis can be extended to the alternating estimation (AltEst) procedure [10] that need not optimize a joint objective over $\mathbf{a}$ and $\mathbf{b}$ and certainly cannot be handled by the earlier strategies.

In order to get (10), the analysis for each AltMin step is often confronted with a technical challenge due to the dependency between data and the iterates obtained so far, which is bypassed by many existing analyses via the resampling assumption. Essentially the resampling-based result states that with any fixed $\hat{\mathbf{b}}_{(t)}$ ($\hat{\mathbf{a}}_{(t+1)}$), given a fresh sample $\mathcal{D}_{(t)}$ independent of $\hat{\mathbf{b}}_{(t)}$ ($\hat{\mathbf{a}}_{(t+1)}$), the next iterate $\hat{\mathbf{a}}_{(t+1)}$ ($\hat{\mathbf{b}}_{(t+1)}$) satisfies the corresponding bound in (10) with high probability. To avoid the resampling, we leverage the idea of uniform bounds [40], which aims to show that given a sample $\mathcal{D}$, the bounds in (10) hold *uniformly* with high probability for *all* possible value of the input $\hat{\mathbf{b}}_{(t)}$ and $\hat{\mathbf{a}}_{(t+1)}$. This argument asks for no fresh data in each iteration, and the probability of the error bounds being true does not deteriorate with growing number of iterations. For structured multi-response regression, we will focus on the AltMin procedure shown in Algorithm 1. For the rest of the paper, $C_0, C_1, c_0, c_1$ and so on are reserved for absolute constants.

## 3   Statistical Guarantees of Alternating Minimization

In this section, we apply the resampling-free analysis strategy introduced in Section 2 to the multi-response regression problem, for which $\mathbf{a} = \boldsymbol{\Sigma}$ and $\mathbf{b} = \boldsymbol{\theta}$. First we introduce a few notations. Given a set $\mathcal{A} \subseteq \mathbb{R}^p$, define $\operatorname{cone} \mathcal{A} = \{c \cdot \mathbf{a} \mid c \geq 0,\ \mathbf{a} \in \mathcal{A}\}$. We denote the smallest and the largest eigenvalue of $\boldsymbol{\Sigma}_*$ as $\sigma_*^-$ and $\sigma_*^+$, and assume $\operatorname{Diag}(\boldsymbol{\Sigma}_*) = \mathbf{I}_{m \times m}$ throughout the paper for simplicity. In addition, we drop the subscripts indexing the iteration, and analyze both $\boldsymbol{\Sigma}$-update and $\boldsymbol{\theta}$-update in

---

**Algorithm 1** Alternating minimization for multi-response regression

---

**Input:** Number of iterations $T$, Data $\mathcal{D} = \{(\mathbf{X}_i, \mathbf{y}_i)\}_{i=1}^n$ and Tuning parameter $\lambda$
**Output:** Estimated $\hat{\boldsymbol{\theta}}_{(T)}$

1: Initialize $\hat{\boldsymbol{\theta}}_{(0)}$ (e.g., solving (4) with $\hat{\boldsymbol{\Sigma}}_{(0)} = \mathbf{I}$)
2: **for** $t := 0$ to $T - 1$ **do**
3:     Compute $\hat{\boldsymbol{\Sigma}}_{(t+1)}$ according to (3)
4:     Compute $\hat{\boldsymbol{\theta}}_{(t+1)}$ by solving (4)
5: **end for**
6: **return** $\hat{\boldsymbol{\theta}}_{(T)}$

---

a broader setting, where the other parameter is fixed as a generic input in certain regions, i.e.,

$$\hat{\boldsymbol{\Sigma}}(\boldsymbol{\theta}) = \frac{1}{n} \sum_{i=1}^n \left(\mathbf{y}_i - \mathbf{X}_i \boldsymbol{\theta}\right) \left(\mathbf{y}_i - \mathbf{X}_i \boldsymbol{\theta}\right)^T, \tag{13}$$

$$\hat{\boldsymbol{\theta}}(\boldsymbol{\Sigma}) = \underset{f(\boldsymbol{\theta}) \leq f(\boldsymbol{\theta}_*)}{\operatorname{argmin}} \frac{1}{2n} \sum_{i=1}^n \left\|\boldsymbol{\Sigma}^{-\frac{1}{2}} \left(\mathbf{y}_i - \mathbf{X}_i \boldsymbol{\theta}\right)\right\|_2^2. \tag{14}$$

Note that here the tuning parameter $\lambda$ in (4) for the $\boldsymbol{\theta}$-step is set as $\lambda = f(\boldsymbol{\theta}_*)$, which will be kept for the rest of the analysis. Given the recent progress in non-convex optimization [3], we also assume that $\hat{\boldsymbol{\theta}}(\boldsymbol{\Sigma})$ can be solved globally despite the potential non-convexity of $f$. The input regions we consider for $\boldsymbol{\theta}$ and $\boldsymbol{\Sigma}$ are respectively given by

$$\mathcal{R} = \left\{\boldsymbol{\theta} \in \mathbb{R}^p \mid f(\boldsymbol{\theta}) \leq f(\boldsymbol{\theta}_*)\right\}, \tag{15}$$

$$\mathcal{M}(e_0) = \left\{\hat{\boldsymbol{\Sigma}}(\boldsymbol{\theta}) \in \mathbb{R}^{m \times m} \mid \boldsymbol{\theta} \in \mathcal{R}, \|\boldsymbol{\theta} - \boldsymbol{\theta}_*\|_2 \leq e_0\right\}, \tag{16}$$

in which $e_0$ is the error tolerance to be specified for the initialization. Note that the input region $\mathcal{M}(e_0)$ implicitly depends on $\mathcal{R}$ as well as the sample $\mathcal{D} = \{(\mathbf{x}, \mathbf{y})\}_{i=1}^n$ used for computing $\hat{\boldsymbol{\Sigma}}(\boldsymbol{\theta})$.

## 3.1 Preliminaries

To apply the proof strategy for AltMin, we first define the distance function $d_1$ and $d_2$.

**Definition 1 (distance functions)** The distance functions for $\boldsymbol{\Sigma}$-step and $\boldsymbol{\theta}$-step are defined as

$$d_1(\boldsymbol{\Sigma}, \boldsymbol{\Sigma}_*) = \frac{\xi(\boldsymbol{\Sigma})}{\xi(\boldsymbol{\Sigma}_*)} - 1, \text{ where } \xi(\boldsymbol{\Sigma}) = \frac{\sqrt{\operatorname{Tr}(\boldsymbol{\Sigma}^{-1} \boldsymbol{\Sigma}_* \boldsymbol{\Sigma}^{-1})}}{\operatorname{Tr}(\boldsymbol{\Sigma}^{-1})}, \tag{17}$$

$$d_2(\boldsymbol{\theta}, \boldsymbol{\theta}_*) = \|\boldsymbol{\theta} - \boldsymbol{\theta}_*\|_2. \tag{18}$$

Although $d_1$ may look odd at first glance, it actually arises as a natural choice after we fix $d_2$, as the $L_2$-error of $\boldsymbol{\theta}$ is our primary goal in the statistical analysis. It is worth noting that $\xi(\boldsymbol{\Sigma})$ is minimized at $\boldsymbol{\Sigma} = \boldsymbol{\Sigma}_*$. The following definition is critical to the analysis for general structures of $\boldsymbol{\theta}_*$ [7].

**Definition 2 (error spherical cap)** For a structure-inducing $f$, its error spherical cap is defined as

$$\mathcal{C} = \operatorname{cone}\left\{\mathbf{u} \in \mathbb{R}^p \mid f(\boldsymbol{\theta}_* + \mathbf{u}) \leq f(\boldsymbol{\theta}_*)\right\} \cap \mathbb{S}^{p-1}, \tag{19}$$

where $\mathbb{S}^{p-1} = \{\mathbf{u} \mid \|\mathbf{u}\|_2 = 1\}$ is the unit sphere of $\mathbb{R}^p$.

The probabilistic analysis of $d_1$ and $d_2$ is built upon the concept of sub-Gaussian vectors and matrices, which are defined below.

**Definition 3 (sub-Gaussian vector and matrix)** A vector $\mathbf{x} \in \mathbb{R}^p$ is said to be sub-Gaussian if its $\psi_2$-norm satisfies,

$$\|\mathbf{x}\|_{\psi_2} = \sup_{\mathbf{u} \in \mathbb{S}^{p-1}} \||\langle \mathbf{x}, \mathbf{u} \rangle|\|_{\psi_2} \leq \kappa < +\infty, \tag{20}$$

where $\|\cdot\|_{\psi_2}$ is defined for a random variable $x \in \mathbb{R}$ as $\|x\|_{\psi_2} = \sup_{q \geq 1} \frac{(\mathbb{E}|x|^q)^{\frac{1}{q}}}{\sqrt{q}}$ . A matrix $\mathbf{X} \in \mathbb{R}^{m \times p}$ is sub-Gaussian if the following $\psi_2$-norm for $\mathbf{X}$ is finite,

$$\|\mathbf{X}\|_{\psi_2} = \sup_{\mathbf{u} \in \mathbb{S}^{p-1}} \sup_{\mathbf{v} \in \mathbb{S}^{m-1}} \left\| \mathbf{u}^T \boldsymbol{\Gamma}_{\mathbf{v}}^{-\frac{1}{2}} \mathbf{X}^T \mathbf{v} \right\|_{\psi_2} \leq \kappa < +\infty , \tag{21}$$

where $\boldsymbol{\Gamma}_{\mathbf{v}} = \mathbb{E}[\mathbf{X}^T \mathbf{v} \mathbf{v}^T \mathbf{X}]$. Further, $\boldsymbol{\Gamma}_{\mathbf{v}}$ for any $\mathbf{v} \in \mathbb{S}^{m-1}$ is assumed to satisfy the condition $0 < \mu^- \leq \lambda_{\min}(\boldsymbol{\Gamma}_{\mathbf{v}}) \leq \lambda_{\max}(\boldsymbol{\Gamma}_{\mathbf{v}}) \leq \mu^+ < +\infty$, for some constants $\mu^-$ and $\mu^+$.

This definition is adopted from [41, 20]. If rows of $\mathbf{X}$ are i.i.d. copies of an isotropic sub-Gaussian random vector $\mathbf{x}$ with $\|\mathbf{x}\|_{\psi_2} \leq \kappa$, it is not difficult to verify that $\|\mathbf{X}\|_{\psi_2} \leq C\kappa$ for a universal constant $C$, and $\mu^- = \mu^+ = 1$. Our assumptions on $\{\mathbf{X}_i\}$ and $\{\tilde{\boldsymbol{\eta}}_i\}$ are given below.

**(A1)** The designs $\mathbf{X}_1, \ldots, \mathbf{X}_n$ are i.i.d. copies of a sub-Gaussian $\mathbf{X}$ with parameter $\kappa$, $\mu^-$ and $\mu^+$.

**(A2)** The isotropic noises $\tilde{\boldsymbol{\eta}}_1, \ldots, \tilde{\boldsymbol{\eta}}_n$ are i.i.d. copies of a sub-Gaussian $\tilde{\boldsymbol{\eta}}$ with parameter $\tau$.

Another key ingredient in the analysis is the complexity measure of the parameter structure captured by $\mathcal{C}$, which turns out to be the notion of Gaussian width [16].

**Definition 4 (Gaussian width)** The Gaussian width $w(\mathcal{A})$ of a set $\mathcal{A} \subseteq \mathbb{R}^p$ is defined as

$$w(\mathcal{A}) = \mathbb{E}_{\mathbf{g} \sim \mathcal{N}(\mathbf{0}, \mathbf{I})} \left[ \sup_{\mathbf{u} \in \mathcal{A}} \langle \mathbf{g}, \mathbf{u} \rangle \right] . \tag{22}$$

Gaussian width is easy to calculate or bound for the error spherical caps induced by many $f$ of interest [7, 9]. Based on Gaussian width, the proofs of the error bounds utilize a powerful tool from probability theory, called generic chaining [37]. We refer the interested readers to the recent monograph [38] and references therein.

### 3.2 Error Bound for Arbitrary Initializations

Given the definitions of distance function $d_1$ and $d_2$, we first focus on the separate error bounds for the $\boldsymbol{\Sigma}$-step and the $\boldsymbol{\theta}$-step in (13) and (14). To allow arbitrary initializations, we consider the tolerance of initialization error $e_0 = +\infty$, which appears in the definition of $\mathcal{M}(e_0)$.

**Lemma 1 (error bound for $\boldsymbol{\Sigma}$-estimation)** *Under the assumptions (A1) and (A2), if the sample size* $n \geq C_0 \max \left\{ 1, \tau^4, \kappa^4 \left( \frac{\sigma_*^+ \mu^+}{\sigma_*^- \mu^-} \right)^2 \right\} \cdot \max \left\{ m, \frac{w^4(\mathcal{C})}{m} \right\}$, *with probability at least* $1 - C_2 \exp(-C_1 m)$, $\hat{\boldsymbol{\Sigma}}(\boldsymbol{\theta})$ *given in (13) is invertible for any* $\boldsymbol{\theta} \in \mathcal{R}$ *and its error satisfies*

$$d_1 \left( \hat{\boldsymbol{\Sigma}}(\boldsymbol{\theta}), \ \boldsymbol{\Sigma}_* \right) \leq C_3 \tau^2 \sqrt{\frac{m}{n}} + C_4 \sqrt{\frac{\mu^+}{\sigma_*^-}} \cdot d_2 \left( \boldsymbol{\theta}, \boldsymbol{\theta}_* \right) . \tag{23}$$

**Remark:** If $\boldsymbol{\theta} = \boldsymbol{\theta}_*$, the $\boldsymbol{\Sigma}$-step computes the sample covariance of the noise, for which $d_2(\boldsymbol{\theta}, \boldsymbol{\theta}_*) = 0$, and the remaining $O\left(\sqrt{\frac{m}{n}}\right)$ term in (23) is the typical statistical rate for covariance estimation.

**Lemma 2 (error bound for $\boldsymbol{\theta}$-estimation)** *Under the assumptions (A1) and (A2), if the sample size* $n \geq C_0 \max \left\{ 1, \tau^4, \kappa^4 \left( \frac{\sigma_*^+ \mu^+}{\sigma_*^- \mu^-} \right)^2 \right\} \cdot \max \left\{ m, \frac{w^4(\mathcal{C})}{m} \right\}$, *then with probability at least* $1 - C_2 \exp(-C_1 m)$, *the following bound holds for* $\hat{\boldsymbol{\theta}}(\boldsymbol{\Sigma})$ *given in (14) with any input* $\boldsymbol{\Sigma} \in \mathcal{M}(+\infty)$,

$$d_2 \left( \hat{\boldsymbol{\theta}}(\boldsymbol{\Sigma}), \ \boldsymbol{\theta}_* \right) \leq (1 + d_1 (\boldsymbol{\Sigma}, \boldsymbol{\Sigma}_*)) \cdot \frac{C_4 \kappa \sqrt{\mu^+}}{\mu^- \sqrt{\mathrm{Tr}(\boldsymbol{\Sigma}_*^{-1})}} \cdot \frac{m + w(\mathcal{C})}{\sqrt{n}} , \tag{24}$$

*where* $\xi(\boldsymbol{\Sigma})$ *is given in Definition 1.*

**Remark:** For $\boldsymbol{\Sigma} = \boldsymbol{\Sigma}_*$ and $\boldsymbol{\Sigma} = \mathbf{I}$, $\boldsymbol{\theta}$-step corresponds to the oracle estimator $\hat{\boldsymbol{\theta}}_{\mathrm{orc}}$ and the ordinary least squares (OLS) estimator $\hat{\boldsymbol{\theta}}_{\mathrm{odn}}$ respectively, i.e.,

$$\hat{\boldsymbol{\theta}}_{\mathrm{orc}} = \underset{f(\boldsymbol{\theta}) \leq f(\boldsymbol{\theta}_*)}{\arg\min} \frac{1}{2n} \sum_{i=1}^{n} \left\| \boldsymbol{\Sigma}_*^{-\frac{1}{2}} (\mathbf{y}_i - \mathbf{X}_i \boldsymbol{\theta}) \right\|_2^2, \tag{25}$$

$$\hat{\boldsymbol{\theta}}_{\mathrm{odn}} = \underset{f(\boldsymbol{\theta}) \leq f(\boldsymbol{\theta}_*)}{\arg\min} \frac{1}{2n} \sum_{i=1}^{n} \left\| \mathbf{y}_i - \mathbf{X}_i \boldsymbol{\theta} \right\|_2^2. \tag{26}$$

An analysis similar to [10] shows that with high probability the $L_2$-errors of $\hat{\boldsymbol{\theta}}_{\mathrm{orc}}$ and $\hat{\boldsymbol{\theta}}_{\mathrm{odn}}$ satisfy

$$\left\| \hat{\boldsymbol{\theta}}_{\mathrm{orc}} - \boldsymbol{\theta}_* \right\|_2 \leq \frac{C' \kappa \sqrt{\mu^+}}{\mu^- \sqrt{\mathrm{Tr}(\boldsymbol{\Sigma}_*^{-1})}} \cdot \frac{w(\mathcal{C})}{\sqrt{n}} \triangleq e_{\mathrm{orc}}, \tag{27}$$

$$\left\| \hat{\boldsymbol{\theta}}_{\mathrm{odn}} - \boldsymbol{\theta}_* \right\|_2 \leq \frac{C' \kappa \sqrt{\mu^+}}{\mu^- \sqrt{m}} \cdot \frac{w(\mathcal{C})}{\sqrt{n}} \triangleq e_{\mathrm{odn}}, \tag{28}$$

which indicates that the oracle estimator improves the OLS by a factor of

$$\frac{e_{\mathrm{orc}}}{e_{\mathrm{odn}}} = \sqrt{\frac{m}{\mathrm{Tr}(\boldsymbol{\Sigma}_*^{-1})}}. \tag{29}$$

In practice, this improvement can be significant, especially when there is strong cross-correlation among the responses, such that $\boldsymbol{\Sigma}_*$ is close to singular.

By assembling Lemma 1 and 2, we obtain the following theorem for the error of AltMin, which exhibits a geometrical convergence to certain *minimum achievable error*.

**Theorem 1 (error bound for arbitrarily-initialized AltMin)** *Under the assumptions (A1) and (A2), if the sample size* $n \geq C_0 \cdot \max \left\{ 1, \tau^4, \kappa^4 \left( \frac{\mu^+ \sigma_*^+}{\mu^- \sigma_*^-} \right)^2, \kappa^2 \left( \frac{\mu^+}{\mu^-} \right)^2 \left( \frac{\sigma_*^+}{\sigma_*^-} \right) \right\} \cdot \max \left\{ \frac{w^4(\mathcal{C})}{m}, m \right\}$, *and* $\hat{\boldsymbol{\theta}}_{(0)}$ *is a feasible initialization (i.e.,* $f(\hat{\boldsymbol{\theta}}_{(0)}) \leq f(\boldsymbol{\theta}_*)$), *then with probability at least* $1 - C_2 \exp(-C_1 m)$, *the following error bound holds for* $\hat{\boldsymbol{\theta}}_{(T)}$ *returned by Algorithm 1*

$$\left\| \hat{\boldsymbol{\theta}}_{(T)} - \boldsymbol{\theta}_* \right\|_2 \leq e_{\min} + \rho_n^T \cdot \left( \left\| \hat{\boldsymbol{\theta}}_{(0)} - \boldsymbol{\theta}_* \right\|_2 - e_{\min} \right), \tag{30}$$

*in which* $\rho_n$ *and* $e_{\min}$ *satisfy the inequalities below with* $\delta_n = C_5 \tau^2 \sqrt{\frac{m}{n}} \leq \frac{1}{4}$,

$$\rho_n \leq \frac{C_3 \kappa \mu^+}{\mu^- \sqrt{\sigma_*^- \mathrm{Tr}(\boldsymbol{\Sigma}_*^{-1})}} \cdot \frac{m + w(\mathcal{C})}{\sqrt{n}} \leq \frac{1}{2}, \tag{31}$$

$$e_{\min} \leq \frac{C_4 \kappa \sqrt{\mu^+}}{\mu^- \sqrt{\mathrm{Tr}(\boldsymbol{\Sigma}_*^{-1})}} \cdot \frac{m + w(\mathcal{C})}{\sqrt{n}} \cdot \frac{1 + \delta_n}{1 - \rho_n}. \tag{32}$$

**Remark:** The inequality (30) indicates that the upper bound of the error for AltMin procedure will decrease *geometrically* to the minimum achievable error $e_{\min}$ with rate $\rho_n$. Though the initialization condition $f(\hat{\boldsymbol{\theta}}_{(0)}) \leq f(\boldsymbol{\theta}_*)$ may not be true for arbitrary $\hat{\boldsymbol{\theta}}_{(0)}$, it should be satisfied by the first iterate $\hat{\boldsymbol{\theta}}_{(1)}$, from which Theorem 1 starts to apply.

Note that the $\rho_n$ in (30) not only controls the convergence rate of error, but also affects the value of $e_{\min}$. The $e_{\min}$ is of the same order as the right-hand side of (24) with $\boldsymbol{\Sigma} = \boldsymbol{\Sigma}_*$, which has an extra additive $O\left( \frac{m}{\sqrt{n}} \right)$ term compared with $e_{\mathrm{orc}}$. This is due to the uniformity considered for the $\boldsymbol{\theta}$-step over all $\boldsymbol{\Sigma} \in \mathcal{M}(+\infty)$. To improve the bound for AltMin, we can consider a small $e_0$ for $\mathcal{M}(e_0)$.

### 3.3 Improved Bound with Good Initializations

As discussed above, we consider a smaller input region $\mathcal{M}(e_0)$ for the $\boldsymbol{\theta}$-step with $e_0 = \sqrt{\frac{\sigma_*^-}{\mu^+}}$. Before presenting the results, we introduce the set called *error spherical sector*.

**Definition 5 (error spherical sector)** For a structure-inducing $f$, its error spherical sector is defined as

$$\mathcal{S} = \text{cone} \left\{ \mathbf{u} \in \mathbb{R}^p \mid f(\boldsymbol{\theta}_* + \mathbf{u}) \le f(\boldsymbol{\theta}_*) \right\} \cap \mathbb{B}^{p-1} , \tag{33}$$

where $\mathbb{B}^p = \{ \mathbf{u} \mid \|\mathbf{u}\|_2 \le 1 \}$ is the unit ball of $\mathbb{R}^p$.

Geometrically $\mathcal{S}$ is closely related to the previously defined $\mathcal{C}$ in (19), and their Gaussian widths satisfy $w(\mathcal{S}) \le w(\mathcal{C}) + c$ for some universal constant $c$. Based on this definition, the following theorem characterizes the sharpened error of AltMin under good initializations.

**Theorem 2 (error bound for well-initialized AltMin)** *Under the assumptions (A1) and (A2), if the sample size* $n \ge C_0 \cdot \max \left\{ 1, \tau^4, \kappa^4 \left( \frac{\mu^+ \sigma_*^+}{\mu^- \sigma_*^-} \right)^2, \kappa^2 \left( \frac{\mu^+}{\mu^-} \right)^2 \left( \frac{\sigma_*^+}{\sigma_*^-} \right) \right\} \cdot \max \left\{ \frac{w^4(\mathcal{C})}{m}, \frac{m^3}{w^2(\mathcal{C})}, m^2 \right\},$

*and a feasible initialization* $\hat{\boldsymbol{\theta}}_{(0)}$ *satisfies* $\|\hat{\boldsymbol{\theta}}_{(0)} - \boldsymbol{\theta}_*\|_2 \le \sqrt{\frac{\sigma_*^-}{\mu^+}}$, *then with probability at least* $1 - C_2 \exp \left( -C_1 \cdot \min \left\{ w^2(\mathcal{C}), m \right\} \right)$, *the error bound (30) holds for* $\hat{\boldsymbol{\theta}}_{(T)}$ *returned by Algorithm 1 with* $\rho_n$ *and* $e_{\min}$ *satisfying*

$$\rho_n \le \frac{C_3 \kappa \mu^+}{\mu^- \sqrt{\sigma_*^- \, \text{Tr}(\boldsymbol{\Sigma}_*^{-1})}} \cdot \frac{w(\mathcal{S})}{\sqrt{n}} \le \frac{1}{2} , \tag{34}$$

$$e_{\min} \le \frac{C_4 \kappa \sqrt{\mu^+}}{\mu^- \sqrt{\text{Tr}(\boldsymbol{\Sigma}_*^{-1})}} \cdot \frac{w(\mathcal{S})}{\sqrt{n}} \cdot \frac{1 + \delta_n}{1 - \rho_n} , \tag{35}$$

*where* $\delta_n$ *is the same as the one given in Theorem 1.*

**Remark:** Since $w(\mathcal{S})$ only differs from $w(\mathcal{C})$ by a constant, the above error bound matches the order of the oracle error $e_{\text{orc}}$. For instance, if $\boldsymbol{\theta}_*$ is $s$-sparse and $f = \| \cdot \|_0$, then $w(\mathcal{S})$ and $e_{\min}$ satisfy,

$$w(\mathcal{S}) = O \left( \sqrt{s \log p} \right) \qquad \Longrightarrow \qquad e_{\min} = O \left( \sqrt{\frac{s \log p}{n}} \right)$$

The initialization condition is a result of setting a small value of $e_0$, which yields an improved version of Lemma 2 so that we can obtain a better bound in Theorem 2. A reasonably good initialization of $\hat{\boldsymbol{\theta}}_{(0)}$ can be obtained by solving OLS $\hat{\boldsymbol{\theta}}_{\text{odn}}$, whose error bound is given (27). On the other hand, the iterates obtained by running arbitrarily-initialized AltMin may also satisfy the initialization requirements as Theorem 1 guarantees a moderate error. Once the requirements are met during the iteration, the arbitrarily-initialized AltMin can attain this sharper bound as well as the well-initialized.

## 4  Experiments

In this section, we present some experimental results to support our theoretical analysis. Specifically we focus on the sparsity structure of $\boldsymbol{\theta}_*$, and consider $L_0$-cardinality as complexity function $f$. Throughout the experiment, we fix problem dimension $p = 1000$, sparsity level of $\boldsymbol{\theta}_*$ $s = 20$, and number of iterations $T = 10$. Entries of $\mathbf{X}$ of $\tilde{\boldsymbol{\eta}}$ are generated by i.i.d. standard Gaussian, and $\boldsymbol{\theta}_* = [\underbrace{1, \ldots, 1}_{10}, \underbrace{-1, \ldots, -1}_{10}, \underbrace{0, \ldots, 0}_{980}]^T$. $\boldsymbol{\Sigma}_*$ is given as a block diagonal matrix with $\boldsymbol{\Sigma}' = \begin{bmatrix} 1 & a \\ a & 1 \end{bmatrix}$ replicated along the diagonal. All the plots are obtained based on the average over 100 random trials.

First we set $a = 0.9$, $m = 10$, and vary sample size $n$ from 30 to 80. We run the AltMin initialized by both OLS and Gaussian random vector, where $\boldsymbol{\theta}$-step is solved by the hard-thresholding pursuit (HTP) algorithm [11]. The error plots are shown in Figure 1. Second, we fix $m = 10$, and vary the parameter $a$ in $\boldsymbol{\Sigma}_*$ from 0.5 to 0.9 for $n = 30, 40, 50$ and 60. The plots in Figure 2(a) shows the error of AltMin against $a$. As indicated by (29), the improvement of the oracle least squares over the ordinary one is amplified with increasingly large $a$. Figure 2(b) compares the actual ratio of $e_{\text{orc}}$ to $e_{\text{odn}}$ and the suggested one. Finally we fix $a = 0.8$, and the number of responses $m$ ranges from 10 to 18 for $n = 30, 40, 50$ and 60. The results are presented in Figure 2(c) and 2(d).

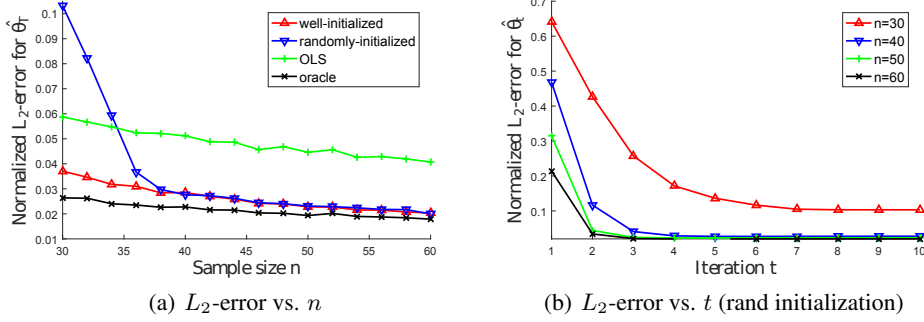

(a) $L_2$-error vs. $n$  (b) $L_2$-error vs. $t$ (rand initialization)

Figure 1: (a) A phase transition is observed for the randomly-initialized AltMin around $n = 40$, whose error is on a par with the well-initialized for $n \geq 40$. This coincides with the remark for Theorem 2. Also, the error of AltMin is close to the oracle estimator, which is significantly better than OLS. (b) Our theoretical results suggest that a larger sample size leads to smaller $\rho_n$, thus AltMin converge faster as shown in the plots.

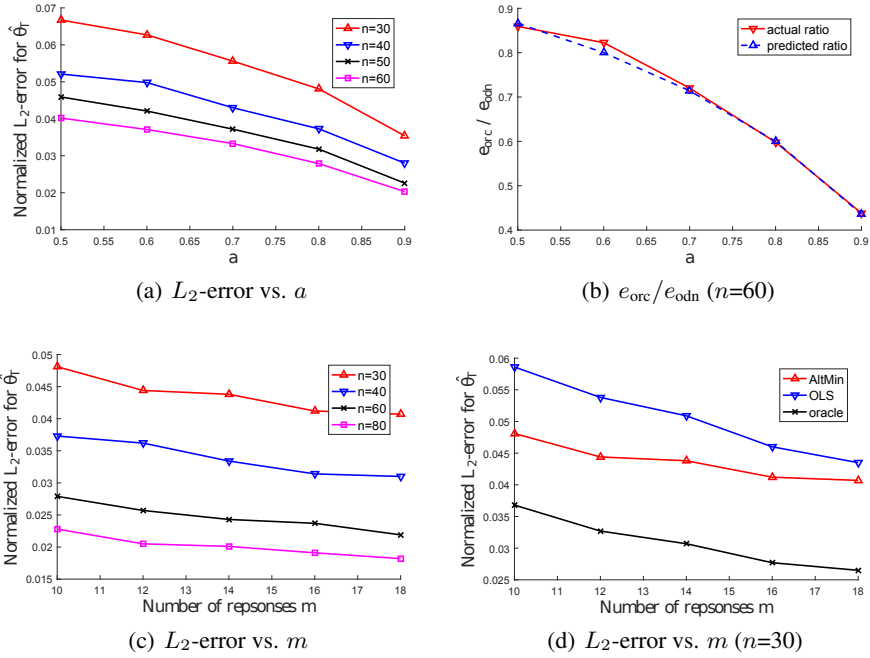

(a) $L_2$-error vs. $a$  (b) $e_{\text{orc}}/e_{\text{odn}}$ ($n$=60)

(c) $L_2$-error vs. $m$  (d) $L_2$-error vs. $m$ ($n$=30)

Figure 2: (a) With $a$ varying from 0.5 to 0.9, the responses become increasingly correlated and the error of AltMin reduces more quickly. (b) The actual ratio of $e_{\text{orc}}$ to $e_{\text{odn}}$ is very close the predicted one given by (29). (c) As $m$ increases from 10 to 18, the error of AltMin does not decrease drastically. The main reason is the increasingly large error in the estimation of $\boldsymbol{\Sigma}_*$. (d) Compared with the error of OLS, the advantage of AltMin becomes marginal with growing $m$, while its gap with the oracle estimator is widened.

## 5 Conclusions

In this paper, we investigate the alternating minimization (AltMin) algorithm for high-dimensional multi-response linear models, which allow general structures of the underlying parameter. In particular, we present a resampling-free analysis for the statistical error of the non-convex AltMin procedure. Our error bound matches the resampling-based result up to some constant, which is of the same order as the oracle estimator. Above all, the error bounds suggest that the arbitrarily-initialized AltMin is able to attain the same level of estimation error as the one with good initializations.

## Acknowledgements

The research was supported by NSF grants IIS-1563950, IIS-1447566, IIS-1447574, IIS-1422557, CCF-1451986, CNS- 1314560, IIS-0953274, IIS-1029711, NASA grant NNX12AQ39A, and gifts from Adobe, IBM, and Yahoo.

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
