[Supplementary Material]

# Supplementary Material to An Improved Analysis of Alternating Minimization for Structured Multi-Response Regression

**Sheng Chen**
The Voleon Group
chen2832@umn.edu

**Arindam Banerjee**
Dept. of Computer Science & Engineering
University of Minnesota, Twin Cities
banerjee@cs.umn.edu

## Abstract

In this supplementary material, we present the proofs of the theoretical results in the main paper. For the ease of exposition, instead of showing the proofs directly, we give a detailed analysis, with several intermediate results included.

## 1 Preliminaries

Our proofs heavily rely on the advanced probability tool, generic chaining [3]. Typically the results in generic chaining are characterized by the so-called $\gamma_2$-functional or its variants [3, 2], whose definitions are complicated. Thanks to the *majorizing measure theorem* (e.g., Theorem 2.4.1 in [4]), we can express those results in terms of Gaussian width, which is sufficient for our purpose. In particular, the following conclusion is adopted from Theorem 2.2.27 in [4].

**Theorem S.1** *Let* $\{Z_{\mathbf{t}}\}_{\mathbf{t} \in \mathcal{T}}$ *be a stochastic process indexed by* $\mathcal{T} \subseteq \mathbb{R}^p$, *which satisfies*

$$\sup_{\mathbf{t}, \mathbf{t}' \in \mathcal{T}} \frac{\||Z_{\mathbf{t}} - Z_{\mathbf{t}'}\||_{\psi_2}}{\|\mathbf{t} - \mathbf{t}'\|_2} \leq K < +\infty \,.$$

*There exist absolute constants* $C_0$ *and* $C_1$ *such that the following bound holds with probability at least* $1 - C_1 \exp\left(-\frac{w^2(\mathcal{T})}{\operatorname{diam}^2(\mathcal{T})}\right)$,

$$\sup_{\mathbf{t}, \mathbf{t}' \in \mathcal{T}} |Z_{\mathbf{t}} - Z_{\mathbf{t}'}| \leq C_0 K \cdot w(\mathcal{T}) \,, \tag{S.1}$$

*where* $\operatorname{diam}(\mathcal{T}) = \sup_{\mathbf{t}, \mathbf{t}' \in \mathcal{T}} \|\mathbf{t} - \mathbf{t}'\|_2$.

In some of the proofs, we also need to bound product processes, which can be handled by the following theorem. This result is essentially a simplified version of Theorem 1.13 in [2]. The original theorem contains a few more tunable variables, which are are not central to the core idea and thus have been hidden.

**Theorem S.2** *Let* $(\Omega, \mu)$ *be a probability space, and* $Z_1, Z_2, \ldots, Z_n$ *be an i.i.d. sample distributed according to* $\mu$. *Suppose that* $\mathcal{F} = \{f_{\mathbf{a}}\}_{\mathbf{a} \in \mathcal{A}}$ *and* $\mathcal{H} = \{h_{\mathbf{b}}\}_{\mathbf{b} \in \mathcal{B}}$ *are two function classes defined on* $(\Omega, \mu)$, *which are indexed by* $\mathcal{A} \subseteq \mathbb{R}^p$ *and* $\mathcal{B} \subseteq \mathbb{R}^q$ *respectively. Assume that*

$$\sup_{f \in \mathcal{F}} \||f\||_{\psi_2} \leq R_{\mathcal{F}} < +\infty \,, \qquad \sup_{h \in \mathcal{H}} \||h\||_{\psi_2} \leq R_{\mathcal{H}} < +\infty \,,$$

$$\sup_{\mathbf{a}, \mathbf{a}' \in \mathcal{A}} \frac{\||f_{\mathbf{a}} - f_{\mathbf{a}'}\||_{\psi_2}}{\|\mathbf{a} - \mathbf{a}'\|_2} \leq K_{\mathcal{F}} < +\infty \,, \qquad \sup_{\mathbf{b}, \mathbf{b}' \in \mathcal{B}} \frac{\||h_{\mathbf{b}} - h_{\mathbf{b}'}\||_{\psi_2}}{\|\mathbf{b} - \mathbf{b}'\|_2} \leq K_{\mathcal{H}} < +\infty \,,$$

*and denote*

$$\varepsilon = \min \left\{ \frac{K_{\mathcal{F}} \cdot w(\mathcal{A})}{R_{\mathcal{F}}}, \ \frac{K_{\mathcal{H}} \cdot w(\mathcal{B})}{R_{\mathcal{H}}} \right\} \ .$$

*There exist absolute constants $C_0$, $C_1$ and $C_2$ such that if $n \geq C_0 \varepsilon^2$, the following inequality holds with probability at least $1 - 2 \exp \left( -C_1 \varepsilon^2 \right)$,*

$$\sup_{f \in \mathcal{F}} \sup_{h \in \mathcal{H}} \left| \frac{1}{n} \sum_{i=1}^{n} f(Z_i) h(Z_i) - \mathbb{E} \left[ fh \right] \right| \ \leq \ C_2 \cdot \frac{R_{\mathcal{H}} K_{\mathcal{F}} \cdot w(\mathcal{A}) + R_{\mathcal{F}} K_{\mathcal{H}} \cdot w(\mathcal{B})}{\sqrt{n}} \tag{S.2}$$

The theorem above immediately leads to the following corollary.

**Corollary S.1** *Under the setting of Theorem S.2, if $\mathcal{F} = \mathcal{H}$ and $\mathcal{A} = \mathcal{B}$, then there exist absolute constants $C_0$, $C_1$ and $C_2$ such that if $n \geq C_0 \left( \frac{K_{\mathcal{F}} \cdot w(\mathcal{A})}{R_{\mathcal{F}}} \right)^2$, the following inequality holds with probability at least $1 - 2 \exp \left( -C_1 \left( \frac{K_{\mathcal{F}} \cdot w(\mathcal{A})}{R_{\mathcal{F}}} \right)^2 \right)$,*

$$\sup_{f \in \mathcal{F}} \left| \frac{1}{n} \sum_{i=1}^{n} f^2(Z_i) - \mathbb{E} \left[ f^2 \right] \right| \ \leq \ C_2 \cdot \frac{R_{\mathcal{F}} K_{\mathcal{F}} \cdot w(\mathcal{A})}{\sqrt{n}} \tag{S.3}$$

The following lemma is also useful in the proof, which essentially states that the concatenation of independent sub-Gaussian random vectors is also sub-Gaussian.

**Lemma S.1** *If $\mathbf{x}_1, \mathbf{x}_2, \ldots, \mathbf{x}_n$ are all $m$-dimensional independent centered sub-Gaussian random vectors, then $\mathbf{x} = [\mathbf{x}_1^T, \ldots, \mathbf{x}_n^T]^T \in \mathbb{R}^{mn}$ is also a centered sub-Gaussian random vector with*

$$\|\mathbf{x}\|_{\psi_2} \leq C \max_{1 \leq i \leq n} \|\mathbf{x}_i\|_{\psi_2} , \tag{S.4}$$

*where $C$ is an absolute constant.*

*Proof:* Define $\mathbf{a} = [\mathbf{a}_1^T, \mathbf{a}_2^T, \ldots, \mathbf{a}_n^T]^T \in \mathbb{S}^{mn-1}$, where each $\mathbf{a}_i$ is $m$-dimensional. We have

$$\|\langle \mathbf{x}, \mathbf{a} \rangle\|_{\psi_2} = \left\| \sum_{i=1}^{n} \langle \mathbf{x}_i, \mathbf{a}_i \rangle \right\|_{\psi_2} \leq \sqrt{C^2 \sum_{i=1}^{n} \|\langle \mathbf{x}_i, \mathbf{a}_i \rangle\|_{\psi_2}^2} \leq \sqrt{C^2 \sum_{i=1}^{n} \|\mathbf{a}_i\|_2^2 \|\mathbf{x}_i\|_{\psi_2}^2}$$

$$\leq \sqrt{C^2 \sum_{i=1}^{n} \|\mathbf{a}_i\|_2^2 \cdot \max_{1 \leq i \leq n} \|\mathbf{x}_i\|_{\psi_2}} = C \max_{1 \leq i \leq n} \|\mathbf{x}_i\|_{\psi_2} ,$$

where we use Lemma 5.9 in [5] for the first inequality. Based on the definition of sub-Gaussian random vector, we complete the proof. ■

Our analysis is organized as follows. In Section 2, we first give the deterministic error bounds for the distance function $d_1$, $d_2$ and the AltMin procedure, under certain conditions. Then we show in Section 3 that those conditions will hold with high probability given our stochastic assumptions. Finally the results in the main paper are directly implied by combining the analysis in Section 2 and 3. Throughout the analysis, $C_0, C_1, c_0, c_1$ and so on are reserved for absolute constants. Standard order notations such as $o(\cdot)$, $O(\cdot)$ and $\Omega(\cdot)$ are used to denote the corresponding growth rates.

## 2 Deterministic Analysis

In this section, we first bound the distance function $d_1$ and $d_2$ defined in Definition 1. We start with a few definitions.

**Definition S.1 (uniformly restricted eigenvalue)** For designs $\mathbf{X}_1, \mathbf{X}_2, \ldots, \mathbf{X}_n$, the smallest uniformly restricted eigenvalue (URE) the for error spherical cap $\mathcal{C} \subseteq \mathbb{S}^{p-1}$ is defined as

$$\alpha_n^- \triangleq \inf_{\mathbf{v} \in \mathbb{S}^{m-1}} \inf_{\mathbf{u} \in \mathcal{C}} \mathbf{u}^T \left( \frac{1}{n} \sum_{i=1}^{n} \mathbf{X}_i^T \mathbf{v}\mathbf{v}^T \mathbf{X}_i \right) \mathbf{u} \tag{S.5}$$

Similarly the largest URE is given as

$$\alpha_n^+ \triangleq \sup_{\mathbf{v} \in \mathbb{S}^{m-1}} \sup_{\mathbf{u} \in \mathcal{C}} \mathbf{u}^T \left( \frac{1}{n} \sum_{i=1}^{n} \mathbf{X}_i^T \mathbf{v}\mathbf{v}^T \mathbf{X}_i \right) \mathbf{u} \tag{S.6}$$

In comparison with the standard restricted eigenvalue [1], the uniformity of the URE is reflected by the infimum and the supremum operation over $\mathbf{v} \in \mathbb{S}^{m-1}$ in the above definitions.

**Definition S.2 (type-I noise-design interaction strength)** For designs $\mathbf{X}_1, \mathbf{X}_2, \ldots, \mathbf{X}_n$ and untransformed noises $\tilde{\boldsymbol{\eta}}_1, \tilde{\boldsymbol{\eta}}_2, \ldots, \tilde{\boldsymbol{\eta}}_n$, the type-I noise-design interaction (NDI) strength is defined as

$$\gamma_n \triangleq \sup_{\mathbf{u} \in \mathcal{C}} \left\| \frac{2}{n} \sum_{i=1}^{n} \mathbf{X}_i \mathbf{u} \tilde{\boldsymbol{\eta}}_i^T \right\|_2 \tag{S.7}$$

**Definition S.3 (type-II noise-design interaction strength)** For designs $\mathbf{X}_1, \mathbf{X}_2, \ldots, \mathbf{X}_n$ and noises $\boldsymbol{\eta}_1, \boldsymbol{\eta}_2, \ldots, \boldsymbol{\eta}_n$, the type-II noise-design interaction (NDI) strength $\beta_n$ for a set of matrices $\mathcal{K}$ is defined as

$$\beta_n(\mathcal{K}) \triangleq \sup_{\boldsymbol{\Sigma} \in \mathcal{K}} \sup_{\mathbf{u} \in \mathcal{C}} \frac{2}{n} \sum_{i=1}^{n} \frac{\boldsymbol{\eta}_i^T \boldsymbol{\Sigma}^{-1} \mathbf{X}_i \mathbf{u}}{\|\boldsymbol{\Sigma}_*^{1/2} \boldsymbol{\Sigma}^{-1}\|_F} , \tag{S.8}$$

where the invertibility is assumed for every $\boldsymbol{\Sigma} \in \mathcal{K}$.

In the analysis, we specifically focus on $\beta_n(\mathcal{M}(e_0))$, as $\mathcal{M}(e_0)$ defined in (16) is the set of input $\boldsymbol{\Sigma}$ under consideration. From its definition, it is not difficult to see that $\beta_n(\mathcal{M}(e_0))$ is a monotonically increasing function of $e_0$, as $\mathcal{M}(e_0) \subseteq \mathcal{M}(e_0')$ for any $e_0 \le e_0'$. In the probabilistic analysis, we will bound $\beta_n(\mathcal{M}(e_0))$ at specific values of $e_0$. With the definitions presented above, we are ready to give the deterministic guarantees for the $\boldsymbol{\Sigma}$-step and the $\boldsymbol{\theta}$-step in (13) and (14) .

**Lemma S.2 (deterministic error bound for $\boldsymbol{\Sigma}$-estimation)** *Given data* $\{(\mathbf{X}_i, \mathbf{y}_i)\}_{i=1}^{n}$, *let* $\{\delta_n\}$ *be a sequence such that*

$$\left\| \frac{1}{n} \sum_{i=1}^{n} \tilde{\boldsymbol{\eta}}_i \tilde{\boldsymbol{\eta}}_i^T - \mathbf{I} \right\|_2 \le \delta_n . \tag{S.9}$$

*If* $\frac{\delta_n \alpha_n^-}{\gamma_n^2} \ge \frac{\sigma_*^+}{4\sigma_*^-}$ *and* $\delta_n \le \frac{1}{4}$, *then* $\hat{\boldsymbol{\Sigma}}(\boldsymbol{\theta})$ *given in (13) is invertible for any* $\boldsymbol{\theta} \in \mathcal{R}$ *and its error satisfies*

$$d_1\left( \hat{\boldsymbol{\Sigma}}(\boldsymbol{\theta}), \ \boldsymbol{\Sigma}_* \right) \le 4\delta_n + 2\sqrt{\frac{\alpha_n^+}{\sigma_*^-}} \cdot d_2\left( \boldsymbol{\theta}, \boldsymbol{\theta}_* \right) . \tag{S.10}$$

*Proof:* We will use the shorthand notation $\hat{\boldsymbol{\Sigma}}$ for $\hat{\boldsymbol{\Sigma}}(\boldsymbol{\theta})$.

$$\frac{\xi(\hat{\boldsymbol{\Sigma}})}{\xi(\boldsymbol{\Sigma}_*)} = \frac{\sqrt{\mathrm{Tr}\left( \hat{\boldsymbol{\Sigma}}^{-1} \boldsymbol{\Sigma}_* \hat{\boldsymbol{\Sigma}}^{-1} \right)}}{\xi(\boldsymbol{\Sigma}_*) \mathrm{Tr}\left( \hat{\boldsymbol{\Sigma}}^{-1} \right)} = \sqrt{\frac{\mathrm{Tr}\left( \boldsymbol{\Sigma}_*^{-1} \right) \cdot \mathrm{Tr}\left( \hat{\boldsymbol{\Sigma}}^{-1} \boldsymbol{\Sigma}_* \hat{\boldsymbol{\Sigma}}^{-1} \right)}{\mathrm{Tr}^2\left( \hat{\boldsymbol{\Sigma}}^{-1} \right)}}$$

$$= \sqrt{\frac{\mathrm{Tr}\left( \hat{\boldsymbol{\Sigma}}^{\frac{1}{2}} \boldsymbol{\Sigma}_*^{-1} \hat{\boldsymbol{\Sigma}}^{\frac{1}{2}} \hat{\boldsymbol{\Sigma}}^{-1} \right) \cdot \mathrm{Tr}\left( \hat{\boldsymbol{\Sigma}}^{-\frac{1}{2}} \boldsymbol{\Sigma}_* \hat{\boldsymbol{\Sigma}}^{-\frac{1}{2}} \hat{\boldsymbol{\Sigma}}^{-1} \right)}{\mathrm{Tr}^2\left( \hat{\boldsymbol{\Sigma}}^{-1} \right)}}$$

$$\leq \sqrt{\frac{\lambda_{\max}\left(\hat{\mathbf{\Sigma}}^{\frac{1}{2}}\mathbf{\Sigma}_*^{-1}\hat{\mathbf{\Sigma}}^{\frac{1}{2}}\right)\operatorname{Tr}\left(\hat{\mathbf{\Sigma}}^{-1}\right)\cdot\lambda_{\max}\left(\hat{\mathbf{\Sigma}}^{-\frac{1}{2}}\mathbf{\Sigma}_*\hat{\mathbf{\Sigma}}^{-\frac{1}{2}}\right)\operatorname{Tr}\left(\hat{\mathbf{\Sigma}}^{-1}\right)}{\operatorname{Tr}^2\left(\hat{\mathbf{\Sigma}}^{-1}\right)}}$$

$$= \sqrt{\lambda_{\max}\left(\hat{\mathbf{\Sigma}}^{\frac{1}{2}}\mathbf{\Sigma}_*^{-1}\hat{\mathbf{\Sigma}}^{\frac{1}{2}}\right)\lambda_{\max}\left(\hat{\mathbf{\Sigma}}^{-\frac{1}{2}}\mathbf{\Sigma}_*\hat{\mathbf{\Sigma}}^{-\frac{1}{2}}\right)} = \sqrt{\frac{\lambda_{\max}\left(\mathbf{\Sigma}_*^{-\frac{1}{2}}\hat{\mathbf{\Sigma}}\mathbf{\Sigma}_*^{-\frac{1}{2}}\right)}{\lambda_{\min}\left(\mathbf{\Sigma}_*^{-\frac{1}{2}}\hat{\mathbf{\Sigma}}\mathbf{\Sigma}_*^{-\frac{1}{2}}\right)}} \, ,$$

where the inequality follows from Von Neumann's trace inequality. Now we try to bound $\lambda_{\max}\left(\mathbf{\Sigma}_*^{-\frac{1}{2}}\hat{\mathbf{\Sigma}}\mathbf{\Sigma}_*^{-\frac{1}{2}}\right)$ and $\lambda_{\min}\left(\mathbf{\Sigma}_*^{-\frac{1}{2}}\hat{\mathbf{\Sigma}}\mathbf{\Sigma}_*^{-\frac{1}{2}}\right)$ separately. Note that any $\boldsymbol{\theta}$ given by the solution of the $\boldsymbol{\theta}$-step in (14) satisfies that $\frac{\boldsymbol{\theta}-\boldsymbol{\theta}_*}{\|\boldsymbol{\theta}-\boldsymbol{\theta}_*\|_2} \in \mathcal{C}$. By the expression for $\hat{\mathbf{\Sigma}}$ in (13), we have for $\lambda_{\max}\left(\mathbf{\Sigma}_*^{-\frac{1}{2}}\hat{\mathbf{\Sigma}}\mathbf{\Sigma}_*^{-\frac{1}{2}}\right)$,

$$\lambda_{\max}\left(\mathbf{\Sigma}_*^{-\frac{1}{2}}\hat{\mathbf{\Sigma}}\mathbf{\Sigma}_*^{-\frac{1}{2}}\right) = 1 + \lambda_{\max}\left(\mathbf{\Sigma}_*^{-\frac{1}{2}}\hat{\mathbf{\Sigma}}\mathbf{\Sigma}_*^{-\frac{1}{2}} - \mathbf{I}\right) = 1 + \left\|\mathbf{\Sigma}_*^{-\frac{1}{2}}\hat{\mathbf{\Sigma}}\mathbf{\Sigma}_*^{-\frac{1}{2}} - \mathbf{I}\right\|_2$$

$$\leq 1 + \left\|\frac{1}{n}\sum_{i=1}^n \tilde{\boldsymbol{\eta}}_i\tilde{\boldsymbol{\eta}}_i^T - \mathbf{I}\right\|_2 + \left\|\frac{2}{n}\sum_{i=1}^n \mathbf{\Sigma}_*^{-\frac{1}{2}}\mathbf{X}_i(\boldsymbol{\theta}-\boldsymbol{\theta}_*)\tilde{\boldsymbol{\eta}}_i^T\right\|_2$$

$$\quad + \lambda_{\max}\left(\frac{1}{n}\sum_{i=1}^n \mathbf{\Sigma}_*^{-\frac{1}{2}}\mathbf{X}_i(\boldsymbol{\theta}-\boldsymbol{\theta}_*)(\boldsymbol{\theta}-\boldsymbol{\theta}_*)^T\mathbf{X}_i^T\mathbf{\Sigma}_*^{-\frac{1}{2}}\right)$$

$$= 1 + \delta_n + \|\boldsymbol{\theta}-\boldsymbol{\theta}_*\|_2 \cdot \left\|\frac{2}{n}\sum_{i=1}^n \mathbf{\Sigma}_*^{-\frac{1}{2}}\mathbf{X}_i \cdot \frac{\boldsymbol{\theta}-\boldsymbol{\theta}_*}{\|\boldsymbol{\theta}-\boldsymbol{\theta}_*\|_2} \cdot \tilde{\boldsymbol{\eta}}_i^T\right\|_2$$

$$\quad + \|\boldsymbol{\theta}-\boldsymbol{\theta}_*\|_2^2 \cdot \sup_{\mathbf{v}\in\mathbb{S}^{m-1}} \mathbf{v}^T\left(\frac{1}{n}\sum_{i=1}^n \mathbf{\Sigma}_*^{-\frac{1}{2}}\mathbf{X}_i \cdot \frac{(\boldsymbol{\theta}-\boldsymbol{\theta}_*)(\boldsymbol{\theta}-\boldsymbol{\theta}_*)^T}{\|\boldsymbol{\theta}-\boldsymbol{\theta}_*\|_2^2} \cdot \mathbf{X}_i^T\mathbf{\Sigma}_*^{-\frac{1}{2}}\right)\mathbf{v}$$

$$\leq 1 + \delta_n + \|\boldsymbol{\theta}-\boldsymbol{\theta}_*\|_2 \cdot \left\|\mathbf{\Sigma}_*^{-\frac{1}{2}}\right\|_2 \cdot \sup_{\mathbf{u}\in\mathcal{C}}\left\|\frac{2}{n}\sum_{i=1}^n \mathbf{X}_i\mathbf{u}\tilde{\boldsymbol{\eta}}_i^T\right\|_2$$

$$\quad + \|\boldsymbol{\theta}-\boldsymbol{\theta}_*\|_2^2 \cdot \left\|\mathbf{\Sigma}_*^{-1}\right\|_2 \cdot \sup_{\mathbf{v}\in\mathbb{S}^{m-1}}\sup_{\mathbf{u}\in\mathcal{C}} \mathbf{u}^T\left(\frac{1}{n}\sum_{i=1}^n \mathbf{X}_i^T\mathbf{v}\mathbf{v}^T\mathbf{X}_i\right)\mathbf{u}$$

$$= 1 + \delta_n + \frac{\gamma_n}{\sqrt{\sigma_*^-}}\|\boldsymbol{\theta}-\boldsymbol{\theta}_*\|_2 + \frac{\alpha_n^+}{\sigma_*^-}\|\boldsymbol{\theta}-\boldsymbol{\theta}_*\|_2^2$$

Similarly we bound $\lambda_{\min}\left(\mathbf{\Sigma}_*^{-\frac{1}{2}}\hat{\mathbf{\Sigma}}\mathbf{\Sigma}_*^{-\frac{1}{2}}\right)$ as follows,

$$\lambda_{\min}\left(\mathbf{\Sigma}_*^{-\frac{1}{2}}\hat{\mathbf{\Sigma}}\mathbf{\Sigma}_*^{-\frac{1}{2}}\right) = 1 + \lambda_{\min}\left(\mathbf{\Sigma}_*^{-\frac{1}{2}}\hat{\mathbf{\Sigma}}\mathbf{\Sigma}_*^{-\frac{1}{2}} - \mathbf{I}\right)$$

$$\geq 1 + \lambda_{\min}\left(\frac{1}{n}\sum_{i=1}^n \tilde{\boldsymbol{\eta}}_i\tilde{\boldsymbol{\eta}}_i^T - \mathbf{I}\right)$$

$$\quad + \lambda_{\min}\left(\frac{1}{n}\sum_{i=1}^n \mathbf{\Sigma}_*^{-\frac{1}{2}}\mathbf{X}_i(\boldsymbol{\theta}-\boldsymbol{\theta}_*)\tilde{\boldsymbol{\eta}}_i^T + \frac{1}{n}\sum_{i=1}^n \tilde{\boldsymbol{\eta}}_i(\boldsymbol{\theta}-\boldsymbol{\theta}_*)^T\mathbf{X}_i^T\mathbf{\Sigma}_*^{-\frac{1}{2}}\right)$$

$$\quad + \lambda_{\min}\left(\frac{1}{n}\sum_{i=1}^n \mathbf{\Sigma}_*^{-\frac{1}{2}}\mathbf{X}_i(\boldsymbol{\theta}-\boldsymbol{\theta}_*)(\boldsymbol{\theta}-\boldsymbol{\theta}_*)^T\mathbf{X}_i^T\mathbf{\Sigma}_*^{-\frac{1}{2}}\right)$$

$$\geq 1 - \left\|\frac{1}{n}\sum_{i=1}^n \tilde{\boldsymbol{\eta}}_i\tilde{\boldsymbol{\eta}}_i^T - \mathbf{I}\right\|_2 - \left\|\frac{2}{n}\sum_{i=1}^n \mathbf{\Sigma}_*^{-\frac{1}{2}}\mathbf{X}_i(\boldsymbol{\theta}-\boldsymbol{\theta}_*)\tilde{\boldsymbol{\eta}}_i^T\right\|_2$$

$$\quad + \lambda_{\min}\left(\frac{1}{n}\sum_{i=1}^n \mathbf{\Sigma}_*^{-\frac{1}{2}}\mathbf{X}_i(\boldsymbol{\theta}-\boldsymbol{\theta}_*)(\boldsymbol{\theta}-\boldsymbol{\theta}_*)^T\mathbf{X}_i^T\mathbf{\Sigma}_*^{-\frac{1}{2}}\right)$$

$$\geq 1 - \delta_n - \|\boldsymbol{\theta} - \boldsymbol{\theta}_*\|_2 \cdot \left\|\boldsymbol{\Sigma}_*^{-\frac{1}{2}}\right\|_2 \cdot \sup_{\mathbf{u} \in \mathcal{C}} \left\|\frac{2}{n}\sum_{i=1}^{n} \mathbf{X}_i \mathbf{u} \tilde{\boldsymbol{\eta}}_i^T\right\|_2$$

$$+ \|\boldsymbol{\theta} - \boldsymbol{\theta}_*\|_2^2 \cdot \lambda_{\min}(\boldsymbol{\Sigma}_*^{-1}) \inf_{\mathbf{v} \in \mathbb{S}^{m-1}} \inf_{\mathbf{u} \in \mathcal{C}} \mathbf{u} \left(\frac{1}{n}\sum_{i=1}^{n} \mathbf{X}_i^T \mathbf{v} \mathbf{v}^T \mathbf{X}_i\right) \mathbf{u}$$

$$= 1 - \delta_n - \frac{\gamma_n}{\sqrt{\sigma_*^-}} \|\boldsymbol{\theta} - \boldsymbol{\theta}_*\|_2 + \frac{\alpha_n^-}{\sigma_*^+} \|\boldsymbol{\theta} - \boldsymbol{\theta}_*\|_2^2$$

Combining the inequalities above, we obtain

$$\frac{\xi(\hat{\boldsymbol{\Sigma}})}{\xi(\boldsymbol{\Sigma}_*)} \leq \sqrt{\frac{1 + \delta_n + \frac{\gamma_n}{\sqrt{\sigma_*^-}}\|\boldsymbol{\theta} - \boldsymbol{\theta}_*\|_2 + \frac{\alpha_n^+}{\sigma_*^-}\|\boldsymbol{\theta} - \boldsymbol{\theta}_*\|_2^2}{1 - \delta_n - \frac{\gamma_n}{\sqrt{\sigma_*^-}}\|\boldsymbol{\theta} - \boldsymbol{\theta}_*\|_2 + \frac{\alpha_n^-}{\sigma_*^+}\|\boldsymbol{\theta} - \boldsymbol{\theta}_*\|_2^2}}$$

$$\leq \sqrt{\frac{1 + 2\delta_n + \frac{\gamma_n^2}{4\sigma_*^- \delta_n}\|\boldsymbol{\theta} - \boldsymbol{\theta}_*\|_2^2 + \frac{\alpha_n^+}{\sigma_*^-}\|\boldsymbol{\theta} - \boldsymbol{\theta}_*\|_2^2}{1 - 2\delta_n - \frac{\gamma_n^2}{4\sigma_*^- \delta_n}\|\boldsymbol{\theta} - \boldsymbol{\theta}_*\|_2^2 + \frac{\alpha_n^-}{\sigma_*^+}\|\boldsymbol{\theta} - \boldsymbol{\theta}_*\|_2^2}} \quad \left(\text{follow from } 2\sqrt{ab} \leq a + b \text{ for } a,b \geq 0\right)$$

$$\leq \sqrt{\frac{1 + 2\delta_n + \frac{2\alpha_n^+}{\sigma_*^-}\|\boldsymbol{\theta} - \boldsymbol{\theta}_*\|_2^2}{1 - 2\delta_n}} \quad \left(\text{use the condition } \frac{\delta_n \alpha_n^-}{\gamma_n^2} \geq \frac{\sigma_*^+}{4\sigma_*^-}\right)$$

$$\leq \sqrt{\frac{1 + 2\delta_n}{1 - 2\delta_n}} + \sqrt{\frac{2\alpha_n^+\|\boldsymbol{\theta} - \boldsymbol{\theta}_*\|_2^2}{(1 - 2\delta_n)\sigma_*^-}} \quad \left(\text{follow from } \sqrt{a+b} \leq \sqrt{a} + \sqrt{b} \text{ for } a,b \geq 0\right)$$

$$\leq 1 + \frac{2\delta_n}{1 - 2\delta_n} + \sqrt{\frac{2\alpha_n^+\|\boldsymbol{\theta} - \boldsymbol{\theta}_*\|_2^2}{(1 - 2\delta_n)\sigma_*^-}} \quad \left(\text{follow from } \sqrt{1+a} \leq 1 + \frac{a}{2} \text{ for } a \geq 0\right)$$

$$\leq 1 + 4\delta_n + 2\sqrt{\frac{\alpha_n^+}{\sigma_*^-}} \cdot \|\boldsymbol{\theta} - \boldsymbol{\theta}_*\|_2 \quad \left(\text{use the condition } \delta_n \leq \frac{1}{4}\right) .$$

The invertibility of $\hat{\boldsymbol{\Sigma}}$ is guaranteed by $\lambda_{\min}(\boldsymbol{\Sigma}_*^{-\frac{1}{2}}\hat{\boldsymbol{\Sigma}}\boldsymbol{\Sigma}_*^{-\frac{1}{2}}) > \frac{1}{2}$ following from the derivation above. ∎

**Lemma S.3 (deterministic error bound for $\theta$-estimation)** *Given data $\{(\mathbf{X}_i, \mathbf{y}_i)\}_{i=1}^n$ and a set $\mathcal{K} \subseteq \mathbb{R}^{m \times m}$ such that every $\boldsymbol{\Sigma} \in \mathcal{K}$ is invertible, if the tuning parameter $\lambda$ is set to $f(\boldsymbol{\theta}_*)$, then the following error bound holds for $\hat{\boldsymbol{\theta}}(\boldsymbol{\Sigma})$ given in (14) with any input $\boldsymbol{\Sigma} \in \mathcal{K}$,*

$$d_2\left(\hat{\boldsymbol{\theta}}(\boldsymbol{\Sigma}), \boldsymbol{\theta}_*\right) \leq \xi(\boldsymbol{\Sigma}) \cdot \frac{\beta_n(\mathcal{K})}{\alpha_n^-} = (1 + d_1(\boldsymbol{\Sigma}, \boldsymbol{\Sigma}_*)) \cdot \xi(\boldsymbol{\Sigma}_*) \cdot \frac{\beta_n(\mathcal{K})}{\alpha_n^-}, \quad \text{(S.11)}$$

*where $\xi(\boldsymbol{\Sigma})$ is defined in Definition 1. In particular, the error for $\hat{\boldsymbol{\theta}}(\boldsymbol{\Sigma})$ with any input $\boldsymbol{\Sigma} \in \mathcal{M}(e_0)$ satisfies*

$$d_2\left(\hat{\boldsymbol{\theta}}(\boldsymbol{\Sigma}), \boldsymbol{\theta}_*\right) \leq \xi(\boldsymbol{\Sigma}) \cdot \frac{\beta_n(\mathcal{M}(e_0))}{\alpha_n^-} . \quad \text{(S.12)}$$

**Remark:** Apart from $\mathcal{K} = \mathcal{M}(e_0)$, other specific instantiations of this lemma also yield interesting error bounds. For example, setting $\mathcal{K} = \{\mathbf{I}\}$ gives us the error for the ordinary least squares $\hat{\boldsymbol{\theta}}_{\text{odn}}$ in (25),

$$\left\|\hat{\boldsymbol{\theta}}_{\text{odn}} - \boldsymbol{\theta}_*\right\|_2 \leq \xi(\mathbf{I}) \cdot \frac{\beta_n(\{\mathbf{I}\})}{\alpha_n^-} = \frac{1}{\sqrt{m}} \cdot \frac{\beta_n(\{\mathbf{I}\})}{\alpha_n^-} \triangleq e_{\text{odn}} . \quad \text{(S.13)}$$

If we choose $\mathcal{K} = \{\boldsymbol{\Sigma}_*\}$, the error bound corresponds to the oracle estimator $\hat{\boldsymbol{\theta}}_{\text{orc}}$,

$$\left\|\hat{\boldsymbol{\theta}}_{\text{orc}} - \boldsymbol{\theta}_*\right\|_2 \leq \xi(\boldsymbol{\Sigma}_*) \cdot \frac{\beta_n(\{\boldsymbol{\Sigma}_*\})}{\alpha_n^-} = \frac{1}{\sqrt{\text{Tr}(\boldsymbol{\Sigma}_*^{-1})}} \cdot \frac{\beta_n(\{\boldsymbol{\Sigma}_*\})}{\alpha_n^-} \triangleq e_{\text{orc}} . \quad \text{(S.14)}$$

*Proof:* We use the shorthand notation $\hat{\boldsymbol{\theta}}$ for $\hat{\boldsymbol{\theta}}(\boldsymbol{\Sigma})$. Since the tuning parameter $\lambda$ is set to $\|\boldsymbol{\theta}_*\|$, the optimality of $\hat{\boldsymbol{\theta}}$ implies that

$$\frac{1}{2n}\sum_{i=1}^{n}\left\|\boldsymbol{\Sigma}^{-\frac{1}{2}}(\mathbf{y}_i - \mathbf{X}_i\hat{\boldsymbol{\theta}})\right\|_2^2 \leq \frac{1}{2n}\sum_{i=1}^{n}\left\|\boldsymbol{\Sigma}^{-\frac{1}{2}}(\mathbf{y}_i - \mathbf{X}_i\boldsymbol{\theta}_*)\right\|_2^2$$

$$\implies \frac{1}{2n}\sum_{i=1}^{n}\left\|\boldsymbol{\Sigma}^{-\frac{1}{2}}(\mathbf{y}_i - \mathbf{X}_i\boldsymbol{\theta}_*) + \boldsymbol{\Sigma}^{-\frac{1}{2}}\mathbf{X}_i(\boldsymbol{\theta}_* - \hat{\boldsymbol{\theta}})\right\|_2^2 \leq \frac{1}{2n}\sum_{i=1}^{n}\left\|\boldsymbol{\Sigma}^{-\frac{1}{2}}(\mathbf{y}_i - \mathbf{X}_i\boldsymbol{\theta}_*)\right\|_2^2$$

$$\implies \frac{1}{2n}\sum_{i=1}^{n}\left\|\boldsymbol{\Sigma}^{-\frac{1}{2}}\mathbf{X}_i(\hat{\boldsymbol{\theta}} - \boldsymbol{\theta}_*)\right\|_2^2 + \frac{1}{n}\sum_{i=1}^{n}(\mathbf{y}_i - \mathbf{X}_i\boldsymbol{\theta}_*)^T\boldsymbol{\Sigma}^{-1}\mathbf{X}_i(\boldsymbol{\theta}_* - \hat{\boldsymbol{\theta}}) \leq 0$$

$$\implies \frac{1}{n}\sum_{i=1}^{n}\left\|\boldsymbol{\Sigma}^{-\frac{1}{2}}\mathbf{X}_i(\hat{\boldsymbol{\theta}} - \boldsymbol{\theta}_*)\right\|_2^2 \leq \frac{2}{n}\sum_{i=1}^{n}\boldsymbol{\eta}_i^T\boldsymbol{\Sigma}^{-1}\mathbf{X}_i(\hat{\boldsymbol{\theta}} - \boldsymbol{\theta}_*)$$

$$\implies \left\|\hat{\boldsymbol{\theta}} - \boldsymbol{\theta}_*\right\|_2 \leq \frac{\frac{2}{n}\sum_{i=1}^{n}\boldsymbol{\eta}_i^T\boldsymbol{\Sigma}^{-1}\mathbf{X}_i \cdot \frac{\hat{\boldsymbol{\theta}} - \boldsymbol{\theta}_*}{\|\hat{\boldsymbol{\theta}} - \boldsymbol{\theta}_*\|_2}}{\frac{1}{n}\sum_{i=1}^{n}\left\|\boldsymbol{\Sigma}^{-\frac{1}{2}}\mathbf{X}_i \cdot \frac{\hat{\boldsymbol{\theta}} - \boldsymbol{\theta}_*}{\|\hat{\boldsymbol{\theta}} - \boldsymbol{\theta}_*\|_2}\right\|_2^2}$$

Now we try to bound the numerator and the denominator on the right-hand side. Note that $f(\hat{\boldsymbol{\theta}}) \leq \lambda = f(\boldsymbol{\theta}_*)$, we thus have $\frac{\hat{\boldsymbol{\theta}} - \boldsymbol{\theta}_*}{\|\hat{\boldsymbol{\theta}} - \boldsymbol{\theta}_*\|_2} \in \mathcal{C}$ according to the definition of the error spherical cap. Assuming the eigenvalue decomposition $\boldsymbol{\Sigma} = \sum_{j=1}^{m}\sigma_j\mathbf{v}_j\mathbf{v}_j^T$, we further get

$$\frac{1}{n}\sum_{i=1}^{n}\left\|\boldsymbol{\Sigma}^{-\frac{1}{2}}\mathbf{X}_i \cdot \frac{\hat{\boldsymbol{\theta}} - \boldsymbol{\theta}_*}{\|\hat{\boldsymbol{\theta}} - \boldsymbol{\theta}_*\|_2}\right\|_2^2 \geq \inf_{\mathbf{u}\in\mathcal{C}}\frac{1}{n}\sum_{i=1}^{n}\left\|\boldsymbol{\Sigma}^{-\frac{1}{2}}\mathbf{X}_i\mathbf{u}\right\|_2^2$$

$$= \inf_{\mathbf{u}\in\mathcal{C}}\frac{1}{n}\sum_{i=1}^{n}\mathbf{u}^T\mathbf{X}_i^T\left(\sum_{j=1}^{m}\sigma_j^{-1}\mathbf{v}_j\mathbf{v}_j^T\right)\mathbf{X}_i\mathbf{u}$$

$$= \inf_{\mathbf{u}\in\mathcal{C}}\sum_{j=1}^{m}\sigma_j^{-1}\cdot\mathbf{u}^T\left(\frac{1}{n}\sum_{i=1}^{n}\mathbf{X}_i^T\mathbf{v}_j\mathbf{v}_j^T\mathbf{X}_i\right)\mathbf{u}$$

$$\geq \left(\sum_{j=1}^{m}\sigma_j^{-1}\right)\cdot\inf_{\mathbf{v}\in\mathbb{S}^{m-1}}\inf_{\mathbf{u}\in\mathcal{C}}\mathbf{u}^T\left(\frac{1}{n}\sum_{i=1}^{n}\mathbf{X}_i^T\mathbf{v}\mathbf{v}^T\mathbf{X}_i\right)\mathbf{u}$$

$$= \alpha_n^- \cdot \mathrm{Tr}(\boldsymbol{\Sigma}^{-1})$$

$$\frac{2}{n}\sum_{i=1}^{n}\boldsymbol{\eta}_i^T\boldsymbol{\Sigma}^{-1}\mathbf{X}_i \cdot \frac{\hat{\boldsymbol{\theta}} - \boldsymbol{\theta}_*}{\|\hat{\boldsymbol{\theta}} - \boldsymbol{\theta}_*\|_2} \leq \sup_{\mathbf{u}\in\mathcal{C}}\frac{2}{n}\sum_{i=1}^{n}\boldsymbol{\eta}_i^T\boldsymbol{\Sigma}^{-1}\mathbf{X}_i\mathbf{u}$$

$$= \left\|\boldsymbol{\Sigma}_*^{1/2}\boldsymbol{\Sigma}^{-1}\right\|_F \cdot \sup_{\mathbf{u}\in\mathcal{C}}\frac{2}{n}\sum_{i=1}^{n}\frac{\boldsymbol{\eta}_i^T\boldsymbol{\Sigma}^{-1}\mathbf{X}_i\mathbf{u}}{\|\boldsymbol{\Sigma}_*^{1/2}\boldsymbol{\Sigma}^{-1}\|_F}$$

$$\leq \left\|\boldsymbol{\Sigma}_*^{1/2}\boldsymbol{\Sigma}^{-1}\right\|_F \cdot \sup_{\boldsymbol{\Sigma}\in\mathcal{M}}\sup_{\mathbf{u}\in\mathcal{C}}\frac{2}{n}\sum_{i=1}^{n}\frac{\boldsymbol{\eta}_i^T\boldsymbol{\Sigma}^{-1}\mathbf{X}_i\mathbf{u}}{\|\boldsymbol{\Sigma}_*^{1/2}\boldsymbol{\Sigma}^{-1}\|_F}$$

$$= \beta_n \cdot \sqrt{\mathrm{Tr}(\boldsymbol{\Sigma}^{-1}\boldsymbol{\Sigma}_*\boldsymbol{\Sigma}^{-1})}$$

Combining the results above, we can get (S.12). ∎

Equipped with the deterministic bounds for both $\boldsymbol{\theta}$- and $\boldsymbol{\Sigma}$-step, we have the following theorem for the whole AltMin procedure.

**Theorem S.3 (deterministic error bound for AltMin)** *Define $\varepsilon_n$, $\rho_n$ and $e_{\min}$ as*

$$\varepsilon_n = \xi(\boldsymbol{\Sigma}_*) \cdot \frac{\beta_n(\mathcal{M}(e_0))}{\alpha_n^-}, \qquad \rho_n = 2\varepsilon_n\sqrt{\frac{\alpha_n^+}{\sigma_*^-}}, \qquad e_{\min} = \varepsilon_n \cdot \frac{1 + 4\delta_n}{1 - \rho_n}$$

in which $\delta_n$ is defined in Lemma S.2. Assume that $e_{\min} < e_0$ and the initialization satisfies both $f(\hat{\boldsymbol{\theta}}_{(0)}) \leq f(\boldsymbol{\theta}_*)$ and $\|\hat{\boldsymbol{\theta}}_{(0)} - \boldsymbol{\theta}_*\|_2 \leq e_0$. Under the conditions of Lemma S.2 and S.3, if $\rho_n < 1$, then $\hat{\boldsymbol{\theta}}_{(T)}$ returned by Algorithm 1 satisfies

$$\left\| \hat{\boldsymbol{\theta}}_{(T)} - \boldsymbol{\theta}_* \right\|_2 \leq e_{\min} + \rho_n^T \cdot (e_0 - e_{\min}) \ , \tag{S.15}$$

**Remark:** Note that $e_{\min}$ is given in a multiplicative form in terms of $\varepsilon_n$, which is similar to the bound for the error $e_{\mathrm{orc}}$ incurred by the oracle estimator. The theorem also reveals the role of $e_0$, which is calibrating the quality of initialization. The better the initialization is, the smaller the error $e_{\min}$ is.

*Proof:* Since the initialization $\hat{\boldsymbol{\theta}}_{(0)}$ satisfies $f(\hat{\boldsymbol{\theta}}_{(0)}) \leq f(\boldsymbol{\theta}_*)$ and $\|\hat{\boldsymbol{\theta}}_{(0)} - \boldsymbol{\theta}_*\|_2 \leq e_0$, we have $\hat{\boldsymbol{\Sigma}}_{(1)} \in \mathcal{M}(e_0)$ by Lemma S.2 and S.3, we have for the first iteration of Algorithm 1,

$$d_1 \left( \hat{\boldsymbol{\Sigma}}_{(1)}, \ \boldsymbol{\Sigma}_* \right) \leq 4\delta_n + 2\sqrt{\frac{\alpha_n^+}{\sigma_*^-}} \cdot d_2 \left( \hat{\boldsymbol{\theta}}_{(0)}, \ \boldsymbol{\theta}_* \right)$$

$$d_2 \left( \hat{\boldsymbol{\theta}}_{(1)}, \ \boldsymbol{\theta}_* \right) \leq \xi(\hat{\boldsymbol{\Sigma}}_{(1)}) \cdot \frac{\beta_n(\mathcal{M}(e_0))}{\alpha_n^-} = \varepsilon_n \cdot \left( 1 + d_1 \left( \hat{\boldsymbol{\Sigma}}_{(1)}, \ \boldsymbol{\Sigma}_* \right) \right)$$

Combining the two inequalities, we obtain the recurrence relation for the error of $\hat{\boldsymbol{\theta}}_{(1)}$ and $\hat{\boldsymbol{\theta}}_{(0)}$,

$$d_2 \left( \hat{\boldsymbol{\theta}}_{(1)}, \ \boldsymbol{\theta}_* \right) \leq \varepsilon_n \cdot \left( 1 + 4\delta_n + 2\sqrt{\frac{\alpha_n^+}{\sigma_*^-}} \cdot d_2 \left( \hat{\boldsymbol{\theta}}_{(0)}, \ \boldsymbol{\theta}_* \right) \right)$$

As $\rho_n < 1$ and $e_{\min} \leq e_0$, we have $d_2(\hat{\boldsymbol{\theta}}_{(1)}, \boldsymbol{\theta}_*) \leq e_0$, thus $\hat{\boldsymbol{\Sigma}}_{(2)} \in \mathcal{M}(e_0)$. By induction, we can recursively apply the result to $t = 2, 3, \ldots, T$,

$$d_2 \left( \hat{\boldsymbol{\theta}}_{(T)}, \ \boldsymbol{\theta}_* \right) \leq q_T, \quad \text{where } q_t = \varepsilon_n \left( 1 + 4\delta_n \right) + 2\varepsilon_n \sqrt{\frac{\alpha_n^+}{\sigma_*^-}} \cdot q_{t-1} \text{ and } q_0 \leq e_0$$

Solving the recurrence of $r_t$, we get

$$\begin{aligned}
q_T &= \frac{\varepsilon_n \left( 1 + 4\delta_n \right)}{1 - 2\varepsilon_n \sqrt{\frac{\alpha_n^+}{\sigma_*^-}}} + \left( 2\varepsilon_n \sqrt{\frac{\alpha_n^+}{\sigma_*^-}} \right)^T \cdot \left( q_0 - \frac{\varepsilon_n \left( 1 + 4\delta_n \right)}{1 - 2\varepsilon_n \sqrt{\frac{\alpha_n^+}{\sigma_*^-}}} \right) \\
&= e_{\min} + \rho_n^T \cdot (q_0 - e_{\min}) \\
&\leq e_{\min} + \rho_n^T \cdot (e_0 - e_{\min}) \ ,
\end{aligned}$$

which completes the proof. $\blacksquare$

## 3 Probabilistic Analysis

In order for the deterministic results to hold nontrivially, we need the conditions stated in Theorem S.3 to be satisfied, and the error $e_{\min}$ to decay with growing sample size. The proposition below translates those requirements into the desired individual growth rates of $\alpha_n^-$, $\alpha_n^+$, $\beta_n$, $\gamma_n$ and $\delta_n$, which need to hold (with high probability) when the randomness of $\mathbf{X}$ and $\tilde{\boldsymbol{\eta}}$ is considered.

**Proposition S.1** *For any fixed $e_0$ and an initialization with $f(\hat{\boldsymbol{\theta}}_{(0)}) \leq f(\boldsymbol{\theta}_*)$ and $\|\hat{\boldsymbol{\theta}}_{(0)} - \boldsymbol{\theta}_*\|_2 \leq e_0$, the error bound (S.15) holds with large enough $n$, and we have $\lim_{n \to +\infty} e_{\min} = 0$, if $\alpha_n^-$, $\alpha_n^+$, $\delta_n$, $\gamma_n$ and $\beta_n(\mathcal{M}(e_0))$ satisfy the following conditions,*

   *(i) The smallest and the largest URE: $\alpha_n^- = \Theta(1)$ and $\alpha_n^+ = \Theta(1)$*

   *(ii) The rate of convergence for $\left\| \frac{1}{n} \sum_{i=1}^n \tilde{\boldsymbol{\eta}}_i \tilde{\boldsymbol{\eta}}_i^T - \mathbf{I} \right\|_2$: $\delta_n = o(1)$*

*(iii)* *The type-I noise-design interaction strength:* $\gamma_n = o(\delta_n^{1/2})$

*(iv)* *The type-II noise-design interaction strength:* $\beta_n(\mathcal{M}(e_0)) = o(1)$

*Proof:* Since $\alpha_n^- = \Theta(1)$ and $\beta_n(\mathcal{M}(e_0)) = o(1)$, we have $\lim_{n\to+\infty} \varepsilon_n = \lim_{n\to+\infty} \xi(\mathbf{\Sigma}_*) \cdot \frac{\beta_n(\mathcal{M}(e_0))}{\alpha_n^-} = 0$. As *(ii)* holds, it follows from that $\delta_n \leq \frac{1}{4}$ when $n$ is large. Due to *(iii)*, the condition $\frac{\delta_n \alpha_n^-}{\gamma_n^2} \geq \frac{\sigma_*^+}{4\sigma_*}$ is true for sufficiently large $n$. Given that $\varepsilon_n = o(1)$ and $\alpha_n^+ = \Theta(1)$, we have $\rho_n = o(1)$. With $\rho_n = o(1)$ and $\delta_n = o(1)$, it is easy to see that $e_{\min} \leq e_0$ for large enough $n$ and $\lim_{n\to+\infty} \frac{e_{\min}}{\varepsilon_n} = 1$, thereby $\lim_{n\to+\infty} e_{\min} = 0$. ∎

For the rest of the section, our goal is to show the high-probability non-asymptotic bounds for $\alpha_n^-$, $\alpha_n^+$, $\delta_n$, $\gamma_n$ and $\beta_n$. As a reminder, the stochastic assumptions given in the main paper are listed below.

**(A1)** The designs $\mathbf{X}_1, \ldots, \mathbf{X}_n$ are i.i.d. copies of a sub-Gaussian $\mathbf{X}$ with parameter $\kappa$, $\mu^-$ and $\mu^+$.

**(A2)** The isotropic noises $\tilde{\boldsymbol{\eta}}_1, \ldots, \tilde{\boldsymbol{\eta}}_n$ are i.i.d. copies of a sub-Gaussian $\tilde{\boldsymbol{\eta}}$ with parameter $\tau$.

### 3.1 Bounding $\alpha_n^-$ and $\alpha_n^+$

The lemma below justifies the claim of the condition *(i)*.

**Lemma S.4** *Under the assumption (A1), if the sample size* $n \geq C_0 \max\left\{\kappa^4 \left(\frac{\mu^+}{\mu^-}\right)^2, 1\right\} \cdot \max\left\{w^2(\mathcal{C}), m\right\}$, *with probability at least* $1 - 2\exp\left(-C_1 \max\left\{w^2(\mathcal{C}), m\right\}\right)$, *the smallest and the largest URE satisfy*

$$\frac{1}{2}\mu^- \leq \alpha_n^- \leq \alpha_n^+ \leq \frac{3}{2}\mu^+ , \tag{S.16}$$

*where* $w(\mathcal{C})$ *is the Gaussian width of the error spherical cap.*

*Proof:* First we have

$$\alpha_n^- = \inf_{\mathbf{v}\in\mathbb{S}^{m-1}} \inf_{\mathbf{u}\in\mathcal{C}} \mathbf{u}^T \left(\frac{1}{n}\sum_{i=1}^n \mathbf{X}_i^T \mathbf{v}\mathbf{v}^T \mathbf{X}_i\right) \mathbf{u}$$

$$\geq \inf_{\mathbf{v}\in\mathbb{S}^{m-1}} \inf_{\mathbf{u}\in\mathcal{C}} \mathbf{u}^T \left(\mathbb{E}\left[\mathbf{X}^T \mathbf{v}\mathbf{v}^T \mathbf{X}\right]\right) \mathbf{u} + \inf_{\mathbf{v}\in\mathbb{S}^{m-1}} \inf_{\mathbf{u}\in\mathcal{C}} \mathbf{u}^T \left(\frac{1}{n}\sum_{i=1}^n \mathbf{X}_i^T \mathbf{v}\mathbf{v}^T \mathbf{X}_i - \mathbb{E}\left[\mathbf{X}^T \mathbf{v}\mathbf{v}^T \mathbf{X}\right]\right) \mathbf{u}$$

$$\geq \inf_{\mathbf{v}\in\mathbb{S}^{m-1}} \inf_{\mathbf{u}\in\mathcal{C}} \mathbf{u}^T \left(\mathbb{E}\left[\mathbf{X}^T \mathbf{v}\mathbf{v}^T \mathbf{X}\right]\right) \mathbf{u} - \sup_{\mathbf{v}\in\mathbb{S}^{m-1}} \sup_{\mathbf{u}\in\mathcal{C}} \left|\frac{1}{n}\sum_{i=1}^n (\mathbf{u}^T \mathbf{X}_i^T \mathbf{v})^2 - \mathbb{E}(\mathbf{u}^T \mathbf{X}^T \mathbf{v})^2\right|$$

$$\geq \mu^- - \sup_{\mathbf{v}\in\mathbb{S}^{m-1}} \sup_{\mathbf{u}\in\mathcal{C}} \left|\frac{1}{n}\sum_{i=1}^n (\mathbf{u}^T \mathbf{X}_i^T \mathbf{v})^2 - \mathbb{E}(\mathbf{u}^T \mathbf{X}^T \mathbf{v})^2\right|$$

$$\alpha_n^+ = \sup_{\mathbf{v}\in\mathbb{S}^{m-1}} \sup_{\mathbf{u}\in\mathcal{C}} \mathbf{u}^T \left(\frac{1}{n}\sum_{i=1}^n \mathbf{X}_i^T \mathbf{v}\mathbf{v}^T \mathbf{X}_i\right) \mathbf{u}$$

$$\leq \sup_{\mathbf{v}\in\mathbb{S}^{m-1}} \sup_{\mathbf{u}\in\mathcal{C}} \mathbf{u}^T \left(\mathbb{E}\left[\mathbf{X}^T \mathbf{v}\mathbf{v}^T \mathbf{X}\right]\right) \mathbf{u} + \sup_{\mathbf{v}\in\mathbb{S}^{m-1}} \sup_{\mathbf{u}\in\mathcal{C}} \mathbf{u}^T \left(\frac{1}{n}\sum_{i=1}^n \mathbf{X}_i^T \mathbf{v}\mathbf{v}^T \mathbf{X}_i - \mathbb{E}\left[\mathbf{X}^T \mathbf{v}\mathbf{v}^T \mathbf{X}\right]\right) \mathbf{u}$$

$$\leq \sup_{\mathbf{v}\in\mathbb{S}^{m-1}} \sup_{\mathbf{u}\in\mathcal{C}} \mathbf{u}^T \left(\mathbb{E}\left[\mathbf{X}^T \mathbf{v}\mathbf{v}^T \mathbf{X}\right]\right) \mathbf{u} + \sup_{\mathbf{v}\in\mathbb{S}^{m-1}} \sup_{\mathbf{u}\in\mathcal{C}} \left|\frac{1}{n}\sum_{i=1}^n (\mathbf{u}^T \mathbf{X}_i^T \mathbf{v})^2 - \mathbb{E}(\mathbf{u}^T \mathbf{X}^T \mathbf{v})^2\right|$$

$$\leq \mu^+ + \sup_{\mathbf{v}\in\mathbb{S}^{m-1}} \sup_{\mathbf{u}\in\mathcal{C}} \left|\frac{1}{n}\sum_{i=1}^n (\mathbf{u}^T \mathbf{X}_i^T \mathbf{v})^2 - \mathbb{E}(\mathbf{u}^T \mathbf{X}^T \mathbf{v})^2\right|$$

Now the goal is to bound $\sup_{\mathbf{v}\in\mathbb{S}^{m-1}}\sup_{\mathbf{u}\in\mathcal{C}}\left|\frac{1}{n}\sum_{i=1}^{n}(\mathbf{u}^T\mathbf{X}_i^T\mathbf{v})^2-\mathbb{E}(\mathbf{u}^T\mathbf{X}^T\mathbf{v})^2\right|$. In order to apply Corollary S.1, we let $\mathcal{A}=\mathbb{S}^{m-1}\times\mathcal{C}\subset\mathbb{R}^{m+p}$, $\mathbf{a}=(\mathbf{v},\mathbf{u})$, and the function class $\mathcal{F}=\left\{f_{\mathbf{a}}=\mathbf{u}^T\mathbf{X}^T\mathbf{v}\right\}_{\mathbf{a}\in\mathcal{A}}$. We then verify the conditions required by Corollary S.1 for $\mathcal{F}$ and $\mathcal{A}$.

$$
\begin{aligned}
\sup_{f\in\mathcal{F}}\|f\|_{\psi_2} &= \sup_{\mathbf{v}\in\mathbb{S}^{m-1}}\sup_{\mathbf{u}\in\mathcal{C}}\left\|\mathbf{u}^T\mathbf{X}^T\mathbf{v}\right\|_{\psi_2}\\
&= \sup_{\mathbf{v}\in\mathbb{S}^{m-1}}\sup_{\mathbf{u}\in\mathcal{C}}\left\|\mathbf{u}^T\Gamma_{\mathbf{v}}^{1/2}\Gamma_{\mathbf{v}}^{-1/2}\mathbf{X}^T\mathbf{v}\right\|_{\psi_2}\\
&\leq \kappa\cdot\sup_{\mathbf{v}\in\mathbb{S}^{m-1}}\sup_{\mathbf{u}\in\mathcal{C}}\left\|\Gamma_{\mathbf{v}}^{1/2}\mathbf{u}\right\|_2\\
&\leq \kappa\cdot\sup_{\mathbf{v}\in\mathbb{S}^{m-1}}\left\|\Gamma_{\mathbf{v}}^{1/2}\right\|_2\leq\kappa\sqrt{\mu^+}\qquad\Longrightarrow\qquad R_{\mathcal{F}}=\kappa\sqrt{\mu^+}
\end{aligned}
$$

$$
\begin{aligned}
\forall\,\mathbf{a},\mathbf{a}'\in\mathcal{A},\;\;\|f_{\mathbf{a}}-f_{\mathbf{a}'}\|_{\psi_2} &= \left\|\mathbf{u}^T\mathbf{X}^T\mathbf{v}-\mathbf{u}'^T\mathbf{X}^T\mathbf{v}'\right\|_{\psi_2}\\
&= \left\|(\mathbf{u}-\mathbf{u}')^T\mathbf{X}^T\mathbf{v}+\mathbf{u}'^T\mathbf{X}^T(\mathbf{v}-\mathbf{v}')\right\|_{\psi_2}\\
&\leq \|\mathbf{u}-\mathbf{u}'\|_2\left\|\frac{(\mathbf{u}-\mathbf{u}')^T}{\|\mathbf{u}-\mathbf{u}'\|_2}\mathbf{X}^T\mathbf{v}\right\|_{\psi_2}+\|\mathbf{v}-\mathbf{v}'\|_2\left\|\mathbf{u}'^T\mathbf{X}^T\frac{(\mathbf{v}-\mathbf{v}')}{\|\mathbf{v}-\mathbf{v}'\|_2}\right\|_{\psi_2}\\
&\leq \kappa\sqrt{\mu^+}\left(\|\mathbf{u}-\mathbf{u}'\|_2+\|\mathbf{v}-\mathbf{v}'\|_2\right)\\
&\leq \sqrt{2}\kappa\sqrt{\mu^+}\cdot\sqrt{\|\mathbf{u}-\mathbf{u}'\|_2^2+\|\mathbf{v}-\mathbf{v}'\|_2^2}\\
&= \sqrt{2}\kappa\sqrt{\mu^+}\|\mathbf{a}-\mathbf{a}'\|_2\qquad\Longrightarrow\qquad K_{\mathcal{F}}=\sqrt{2}\kappa\sqrt{\mu^+}
\end{aligned}
$$

It follows from Corollary S.1 that if $n\geq c_0 w^2(\mathcal{A})$, the following result holds with probability at least $1-2\exp\left(-c_1 w^2(\mathcal{A})\right)$,

$$
\sup_{\mathbf{v}\in\mathbb{S}^{m-1}}\sup_{\mathbf{u}\in\mathcal{C}}\left|\frac{1}{n}\sum_{i=1}^{n}(\mathbf{u}^T\mathbf{X}_i^T\mathbf{v})^2-\mathbb{E}(\mathbf{u}^T\mathbf{X}^T\mathbf{v})^2\right|\;\leq\;c_2\cdot\frac{\kappa^2\mu^+\cdot w(\mathcal{A})}{\sqrt{n}}\tag{S.17}
$$

If $n$ further satisfies $n\geq 4c_2^2\kappa^4\left(\frac{\mu^+}{\mu^-}\right)^2 w^2(\mathcal{A})$, then

$$
\sup_{\mathbf{v}\in\mathbb{S}^{m-1}}\sup_{\mathbf{u}\in\mathcal{C}}\left|\frac{1}{n}\sum_{i=1}^{n}(\mathbf{u}^T\mathbf{X}_i^T\mathbf{v})^2-\mathbb{E}(\mathbf{u}^T\mathbf{X}^T\mathbf{v})^2\right|\;\leq\;\frac{1}{2}\mu^-
$$

$$
\Longrightarrow\quad \alpha_n^-\geq\mu^--\frac{1}{2}\mu^-=\frac{1}{2}\mu^-,\quad \alpha_n^+\leq\mu^++\frac{1}{2}\mu^-\leq\frac{3}{2}\mu^+
$$

Finally we note that

$$
\begin{aligned}
w(\mathcal{A}) &= \mathbb{E}\left[\sup_{\mathbf{a}\in\mathcal{A}}\langle\mathbf{a},\mathbf{g}_{m+p}\rangle\right]=\mathbb{E}\left[\sup_{\mathbf{u}\in\mathbb{S}^{m-1}}\langle\mathbf{u},\mathbf{g}_m\rangle+\sup_{\mathbf{v}\in\mathcal{C}}\langle\mathbf{v},\mathbf{g}_p\rangle\right]\\
&= \mathbb{E}\left[\|\mathbf{g}_m\|_2\right]+w(\mathcal{C})=\Theta\left(\sqrt{m}\right)+w(\mathcal{C})
\end{aligned}
$$

By renaming the constants, we finish the proof. ∎

## 3.2 Bounding $\delta_n$

The condition $(ii)$ is simply implied by the following bound for the convergence of sample covariance matrix, which is a direct result of Lemma 5.36 and Theorem 5.39 in [5].

**Proposition S.2** *Under the assumption (A2), there exist absolute constants $C_0$, $C_1$ and $C_2$ such that if $n\geq C_0\tau^4 m$, the following inequality holds with probability at least $1-2\exp\left(-C_1 m\right)$,*

$$
\left\|\frac{1}{n}\sum_{i=1}^{n}\tilde{\boldsymbol{\eta}}_i\tilde{\boldsymbol{\eta}}_i^T-\mathbf{I}\right\|_2\leq C_2\tau^2\sqrt{\frac{m}{n}}\triangleq\delta_n\tag{S.18}
$$

### 3.3 Bounding $\gamma_n$

Next we show that the rate of $\gamma_n$ also has a $\frac{1}{\sqrt{n}}$-dependence as $\delta_n$, thus implying that $\gamma_n = o(\delta_n^{1/2})$ in the condition $(iii)$.

**Lemma S.5** *Under the assumptions (A1) and (A2), if $n \geq C_0 m$, the following inequality holds with probability at least $1 - 2\exp\left(-C_1 m\right)$ for the type-I NDI strength $\gamma_n$,*

$$\gamma_n \leq C_2 \cdot \frac{\kappa \tau \sqrt{\mu^+}(\sqrt{m} + w(\mathcal{C}))}{\sqrt{n}} . \tag{S.19}$$

*Proof:* First we have

$$\gamma_n = \sup_{\mathbf{u} \in \mathcal{C}} \left\| \frac{2}{n} \sum_{i=1}^{n} \mathbf{X}_i \mathbf{u} \tilde{\boldsymbol{\eta}}_i^T \right\|_2 = 2 \sup_{\mathbf{u} \in \mathcal{C}} \sup_{\mathbf{v} \in \mathbb{S}^{m-1}} \sup_{\mathbf{b} \in \mathbb{S}^{m-1}} \frac{1}{n} \sum_{i=1}^{n} \left(\mathbf{v}^T \mathbf{X}_i \mathbf{u}\right) \left(\tilde{\boldsymbol{\eta}}_i^T \mathbf{b}\right)$$

$$= 2 \sup_{\mathbf{u} \in \mathcal{C}} \sup_{\mathbf{v} \in \mathbb{S}^{m-1}} \sup_{\mathbf{b} \in \mathbb{S}^{m-1}} \left| \frac{1}{n} \sum_{i=1}^{n} \left(\mathbf{v}^T \mathbf{X}_i \mathbf{u}\right) \left(\tilde{\boldsymbol{\eta}}_i^T \mathbf{b}\right) - \mathbb{E}\left[\mathbf{v}^T \mathbf{X} \mathbf{u} \tilde{\boldsymbol{\eta}}^T \mathbf{b}\right] \right|$$

Next we use Theorem S.2 to bound the stochastic process above. Let $\mathcal{A} = \mathbb{S}^{m-1} \times \mathcal{C} \subset \mathbb{R}^{m+p}$, $\mathbf{a} = (\mathbf{v}, \mathbf{u})$ and $\mathcal{B} = \mathbb{S}^{m-1}$. Construct $\mathcal{F} = \{f_\mathbf{a} = \mathbf{v}^T \mathbf{X} \mathbf{u}\}_{\mathbf{a} \in \mathcal{A}}$ and $\mathcal{H} = \{h_\mathbf{b} = \tilde{\boldsymbol{\eta}}^T \mathbf{b}\}_{\mathbf{b} \in \mathcal{B}}$. We start by verifying the assumptions. Note that

$$\sup_{f \in \mathcal{F}} \|\|f\|\|_{\psi_2} = \sup_{\mathbf{u} \in \mathcal{C}} \sup_{\mathbf{v} \in \mathbb{S}^{m-1}} \|\|\mathbf{u}^T \mathbf{X}^T \mathbf{v}\|\|_{\psi_2}$$

$$\leq \sup_{\mathbf{u} \in \mathcal{C}} \sup_{\mathbf{v} \in \mathbb{S}^{m-1}} \|\|\mathbf{u}^T \mathbf{X}^T \mathbf{v}\|\|_{\psi_2}$$

$$= \sup_{\mathbf{u} \in \mathcal{C}} \sup_{\mathbf{v} \in \mathbb{S}^{m-1}} \|\|\mathbf{u}^T \boldsymbol{\Gamma}_\mathbf{v}^{1/2} \boldsymbol{\Gamma}_\mathbf{v}^{-1/2} \mathbf{X}^T \mathbf{v}\|\|_{\psi_2}$$

$$\leq \sup_{\mathbf{u} \in \mathcal{C}} \sup_{\mathbf{v} \in \mathbb{S}^{m-1}} \kappa \left\| \boldsymbol{\Gamma}_\mathbf{v}^{1/2} \mathbf{u} \right\|_2$$

$$\leq \kappa \sqrt{\mu^+} \qquad \Longrightarrow \qquad R_\mathcal{F} = \kappa \sqrt{\mu^+}$$

$$\sup_{h \in \mathcal{H}} \|\|h\|\|_{\psi_2} = \sup_{\mathbf{b} \in \mathbb{S}^{m-1}} \|\|\tilde{\boldsymbol{\eta}}^T \mathbf{b}\|\|_{\psi_2} \leq \tau \qquad \Longrightarrow \qquad R_\mathcal{H} = \tau$$

Similar to the proof for Lemma S.4, we have

$$\forall\, \mathbf{a}, \mathbf{a}' \in \mathcal{A}, \ \|\|f_\mathbf{a} - f_{\mathbf{a}'}\|\|_{\psi_2} = \left\|\left\| \mathbf{v}^T \mathbf{X}^T \boldsymbol{\Sigma}_*^{-1/2} \mathbf{u} - \mathbf{v}'^T \mathbf{X}^T \boldsymbol{\Sigma}_*^{-1/2} \mathbf{u}' \right\|\right\|_{\psi_2}$$

$$\leq \left\|\left\| (\mathbf{v} - \mathbf{v}')^T \mathbf{X}^T \mathbf{u} + \mathbf{v}'^T \mathbf{X}^T (\mathbf{u} - \mathbf{u}') \right\|\right\|_{\psi_2}$$

$$\leq \|\mathbf{v} - \mathbf{v}'\|_2 \left\|\left\| \frac{(\mathbf{v} - \mathbf{v}')^T}{\|\mathbf{v} - \mathbf{v}'\|_2} \mathbf{X}^T \mathbf{u} \right\|\right\|_{\psi_2} + \|\mathbf{u} - \mathbf{u}'\|_2 \left\|\left\| \mathbf{v}'^T \mathbf{X}^T \frac{(\mathbf{u} - \mathbf{u}')}{\|\mathbf{u} - \mathbf{u}'\|_2} \right\|\right\|_{\psi_2}$$

$$\leq \kappa \sqrt{\mu^+} \left( \|\mathbf{v} - \mathbf{v}'\|_2 + \|\mathbf{u} - \mathbf{u}'\|_2 \right)$$

$$\leq \sqrt{2} \kappa \sqrt{\mu^+} \cdot \sqrt{\|\mathbf{v} - \mathbf{v}'\|_2^2 + \|\mathbf{u} - \mathbf{u}'\|_2^2}$$

$$= \sqrt{2} \kappa \sqrt{\mu^+} \|\mathbf{a} - \mathbf{a}'\|_2 \qquad \Longrightarrow \qquad K_\mathcal{F} = \sqrt{2} \kappa \sqrt{\mu^+}$$

$$\forall\, \mathbf{b}, \mathbf{b}' \in \mathcal{B}, \ \|\|h_\mathbf{b} - h_{\mathbf{b}'}\|\|_{\psi_2} = \|\|\tilde{\boldsymbol{\eta}}^T (\mathbf{b} - \mathbf{b}')\|\|_{\psi_2} \leq \tau \|\mathbf{b} - \mathbf{b}'\|_2 \qquad \Longrightarrow \qquad K_\mathcal{H} = \tau$$

By invoking Theorem S.2 and noting that $w(\mathbb{S}^{m-1}) = \Theta(\sqrt{m})$, $w(\mathcal{A}) = w(\mathbb{S}^{m-1}) + w(\mathcal{C}) \geq w(\mathcal{B})$, if $n \geq c_0 m$, we get

$$\gamma_n \leq 2 \sup_{\mathbf{u} \in \mathcal{C}} \sup_{\mathbf{v} \in \mathbb{S}^{m-1}} \sup_{\mathbf{b} \in \mathbb{S}^{m-1}} \left| \frac{1}{n} \sum_{i=1}^{n} \left(\mathbf{v}^T \mathbf{X}_i \mathbf{u}\right) \left(\tilde{\boldsymbol{\eta}}_i^T \mathbf{b}\right) - \mathbb{E}\left[\mathbf{v}^T \mathbf{X} \mathbf{u} \tilde{\boldsymbol{\eta}}^T \mathbf{b}\right] \right|$$

$$\leq c_2 \cdot \kappa \tau \sqrt{\mu^+} \cdot \frac{\sqrt{m} + w(\mathcal{C})}{\sqrt{n}}$$

with probability at least $1 - 2\exp\left(-c_1 m\right)$. The proof is completed by renaming the constants. $\blacksquare$

### 3.4 Bounding $\beta_n(\mathcal{M}(e_0))$

Lastly we verify the condition $(iv)$. Given the statement of Theorem S.3, we first bound $\beta_n(\mathcal{M}(e_0))$ for $e_0 = +\infty$, which allows arbitrary initializations of AltMin.

**Lemma S.6** *Suppose that the conditions of Lemma S.2 are satisfied with probability $1 - \epsilon$ when $n \geq n_0$. Under the assumptions (A1) and (A2), if sample size $n \geq \max\left\{n_0, C_0\tau^4 m\right\}$, the type-II NDI strength for $\mathcal{M}(e_0)$ with $e_0 = +\infty$ satisfies,*

$$\beta_n(\mathcal{M}(e_0)) \leq C_3 \cdot \frac{\kappa\sqrt{\mu^+}\,(m + w(\mathcal{C}))}{\sqrt{n}}\,, \tag{S.20}$$

*with probability at least $1 - \epsilon - C_2\exp\left(-C_1 m\right)$.*

*Proof:* When the conditions of Lemma S.2 is satisfied, the invertibility holds for all $\boldsymbol{\Sigma} \in \mathcal{M}$. using the relation $\boldsymbol{\eta} = \boldsymbol{\Sigma}_*^{1/2}\tilde{\boldsymbol{\eta}}$, we have

$$
\begin{aligned}
\beta_n &= \sup_{\boldsymbol{\Sigma}\in\mathcal{M}}\sup_{\mathbf{u}\in\mathcal{C}} \frac{2}{n}\sum_{i=1}^{n} \frac{\boldsymbol{\eta}_i^T\boldsymbol{\Sigma}^{-1}\mathbf{X}_i\mathbf{u}}{\|\boldsymbol{\Sigma}_*^{1/2}\boldsymbol{\Sigma}^{-1}\|_F} \\
&= \sup_{\boldsymbol{\Sigma}\in\mathcal{M}}\sup_{\mathbf{u}\in\mathcal{C}} \frac{2}{n}\sum_{i=1}^{n} \frac{\tilde{\boldsymbol{\eta}}_i^T\boldsymbol{\Sigma}_*^{1/2}\boldsymbol{\Sigma}^{-1}\mathbf{X}_i\mathbf{u}}{\|\boldsymbol{\Sigma}_*^{1/2}\boldsymbol{\Sigma}^{-1}\|_F} \\
&\leq \underbrace{\sup_{\boldsymbol{\Lambda}\in\mathbb{S}^{m\times m-1}}\sup_{\mathbf{u}\in\mathcal{C}} \frac{2}{n}\sum_{i=1}^{n}\tilde{\boldsymbol{\eta}}_i^T\boldsymbol{\Lambda}\mathbf{X}_i\mathbf{u}}_{\nu_n}
\end{aligned}
$$

Therefore we just need to bound $\nu_n$. Since the design and noise are independent, we will consider their randomness in a sequential fashion. The proof proceeds in two steps. First we show that the noises $\tilde{\boldsymbol{\eta}}_1, \tilde{\boldsymbol{\eta}}_2, \ldots, \tilde{\boldsymbol{\eta}}_n$ will behave "well" with high probability. By the word "well", we mean that the following event is true,

$$\mathcal{E} = \left\{\{\tilde{\boldsymbol{\eta}}_i\} \;\Big|\; \sup_{\boldsymbol{\Lambda}\in\mathbb{S}^{m\times m-1}}\frac{1}{n}\sum_{i=1}^{n}\left\|\boldsymbol{\Lambda}^T\tilde{\boldsymbol{\eta}}_i\right\|_2^2 \leq 2\right\}\,. \tag{S.21}$$

Denoting the columns of $\boldsymbol{\Lambda}$ by $\boldsymbol{\lambda}_1, \boldsymbol{\lambda}_2, \ldots, \boldsymbol{\lambda}_m$, we have

$$
\begin{aligned}
\sup_{\boldsymbol{\Lambda}\in\mathbb{S}^{m\times m-1}}\frac{1}{n}\sum_{i=1}^{n}\left\|\boldsymbol{\Lambda}^T\tilde{\boldsymbol{\eta}}_i\right\|_2^2 &= \sup_{\boldsymbol{\Lambda}\in\mathbb{S}^{m\times m-1}}\frac{1}{n}\sum_{i=1}^{n}\mathrm{Tr}\left(\boldsymbol{\Lambda}^T\tilde{\boldsymbol{\eta}}_i\tilde{\boldsymbol{\eta}}_i^T\boldsymbol{\Lambda}\right) \\
&= \sup_{\boldsymbol{\Lambda}\in\mathbb{S}^{m\times m-1}}\sum_{j=1}^{m}\boldsymbol{\lambda}_j^T\left(\frac{1}{n}\sum_{i=1}^{n}\tilde{\boldsymbol{\eta}}_i\tilde{\boldsymbol{\eta}}_i^T\right)\boldsymbol{\lambda}_j \\
&= \sup_{\boldsymbol{\Lambda}\in\mathbb{S}^{m\times m-1}}\sum_{j=1}^{m}\|\boldsymbol{\lambda}_j\|_2^2\cdot\left\|\frac{1}{n}\sum_{i=1}^{n}\tilde{\boldsymbol{\eta}}_i\tilde{\boldsymbol{\eta}}_i^T\right\|_2 \\
&= \left\|\frac{1}{n}\sum_{i=1}^{n}\tilde{\boldsymbol{\eta}}_i\tilde{\boldsymbol{\eta}}_i^T\right\|_2
\end{aligned}
$$

By Proposition S.2, if $n \geq c_0\tau^4 m$, we have

$$\left\|\frac{1}{n}\sum_{i=1}^{n}\tilde{\boldsymbol{\eta}}_i\tilde{\boldsymbol{\eta}}_i^T\right\|_2 \leq 1 + \left\|\frac{1}{n}\sum_{i=1}^{n}\tilde{\boldsymbol{\eta}}_i\tilde{\boldsymbol{\eta}}_i^T - \mathbf{I}\right\|_2 \leq 2$$

with probability at least $1 - 2\exp\left(-c_1 m\right)$.

Next we consider the randomness of $\mathbf{X}_i$ given that $\tilde{\boldsymbol{\eta}}_i$'s are fixed and $\mathcal{E}$ is true. Construct the stochastic process $\left\{Z_{\mathbf{t}} = \frac{1}{\sqrt{n}}\sum_{i=1}^{n}\tilde{\boldsymbol{\eta}}_i^T\boldsymbol{\Lambda}\mathbf{X}_i\mathbf{u}\right\}_{\mathbf{t}\in\mathcal{T}}$, where $\mathcal{T} = \mathbb{S}^{m\times m-1} \times \mathcal{C} \subset \mathbb{R}^{m\times m+p}$ and $\mathbf{t} = (\mathrm{vec}(\boldsymbol{\Lambda}), \mathbf{u})$. Note that

$$\forall\, \mathbf{t}, \mathbf{t}' \in \mathcal{T}, \quad \|\mathbf{t} - \mathbf{t}'\|_2 = \sqrt{\|\boldsymbol{\Lambda} - \boldsymbol{\Lambda}'\|_F^2 + \|\mathbf{u} - \mathbf{u}'\|_2^2} \leq 2\sqrt{2} \quad\Longrightarrow\quad \mathrm{diam}\left(\mathcal{T}\right) \leq 2\sqrt{2}$$

In order to apply Theorem S.1 to $\{Z_{\mathbf{t}}\}$, we first verify the required condition.

$$\forall \, \mathbf{t}, \mathbf{t}' \in \mathcal{T}, \; \|Z_{\mathbf{t}} - Z_{\mathbf{t}'}\|_{\psi_2} = \left\|\!\left\|\frac{1}{\sqrt{n}}\sum_{i=1}^{n} \tilde{\boldsymbol{\eta}}_i^T \boldsymbol{\Lambda} \mathbf{X}_i \mathbf{u} - \frac{1}{\sqrt{n}}\sum_{i=1}^{n} \tilde{\boldsymbol{\eta}}_i^T \boldsymbol{\Lambda}' \mathbf{X}_i \mathbf{u}'\right\|\!\right\|_{\psi_2}$$

$$\leq \left\|\!\left\|\frac{1}{\sqrt{n}}\sum_{i=1}^{n} \tilde{\boldsymbol{\eta}}_i^T (\boldsymbol{\Lambda} - \boldsymbol{\Lambda}') \mathbf{X}_i \mathbf{u}\right\|\!\right\|_{\psi_2} + \left\|\!\left\|\frac{1}{\sqrt{n}}\sum_{i=1}^{n} \tilde{\boldsymbol{\eta}}_i^T \boldsymbol{\Lambda}' \mathbf{X}_i (\mathbf{u} - \mathbf{u}')\right\|\!\right\|_{\psi_2}$$

$$\overset{(a)}{\leq} c_2 \sqrt{\frac{1}{n}\sum_{i=1}^{n} \|(\boldsymbol{\Lambda} - \boldsymbol{\Lambda}')^T \tilde{\boldsymbol{\eta}}_i\|_2^2} \cdot \sup_{\mathbf{v} \in \mathbb{S}^{m-1}} \left\|\!\left\|\mathbf{v}^T \mathbf{X} \mathbf{u}\right\|\!\right\|_{\psi_2}$$

$$+ c_2 \sqrt{\frac{1}{n}\sum_{i=1}^{n} \|\boldsymbol{\Lambda}'^T \tilde{\boldsymbol{\eta}}_i\|_2^2} \cdot \|\mathbf{u} - \mathbf{u}'\|_2 \cdot \sup_{\mathbf{v} \in \mathbb{S}^{m-1}} \left\|\!\left\|\mathbf{v}^T \mathbf{X} \frac{\mathbf{u} - \mathbf{u}'}{\|\mathbf{u} - \mathbf{u}'\|_2}\right\|\!\right\|_{\psi_2}$$

$$\leq \sqrt{2} c_2 \kappa \sqrt{\mu^+} \left(\|\boldsymbol{\Lambda} - \boldsymbol{\Lambda}'\|_F + \|\mathbf{u} - \mathbf{u}'\|_2\right)$$

$$\leq 2 c_2 \kappa \sqrt{\mu^+} \left\|\begin{bmatrix}\mathrm{vec}(\boldsymbol{\Lambda}) \\ \mathbf{u}\end{bmatrix} - \begin{bmatrix}\mathrm{vec}(\boldsymbol{\Lambda}') \\ \mathbf{u}'\end{bmatrix}\right\|_2 \quad \Longrightarrow \quad K = 2 c_2 \kappa \sqrt{\mu^+},$$

where step (a) follows from Lemma S.1. By Theorem S.1, we have for fixed $\{\tilde{\boldsymbol{\eta}}_i\}$ under event $\mathcal{E}$,

$$\nu_n = \frac{2}{\sqrt{n}} \cdot \sup_{\mathbf{t} \in \mathcal{T}} Z_{\mathbf{t}} = \frac{1}{\sqrt{n}} \cdot \sup_{\mathbf{t},\mathbf{t}' \in \mathcal{T}} |Z_{\mathbf{t}} - Z_{\mathbf{t}'}| \leq c_3 \cdot \frac{\kappa \sqrt{\mu^+} \cdot w(\mathcal{T})}{\sqrt{n}}$$

with probability at least $1 - c_4 \exp\left(-\frac{w^2(\mathcal{T})}{\mathrm{diam}^2(\mathcal{T})}\right) \geq 1 - c_4 \exp\left(-\frac{w^2(\mathcal{T})}{8}\right)$. Now we combine the randomness of $\mathbf{X}_i$ and $\tilde{\boldsymbol{\eta}}_i$, and get

$$\mathbb{P}_{\mathbf{X},\tilde{\boldsymbol{\eta}}}\left(\nu_n \leq c_3 \cdot \frac{\kappa \sqrt{\mu^+} \cdot w(\mathcal{T})}{\sqrt{n}}\right)$$

$$= \int \mathbb{P}_{\mathbf{X}}\left(\nu_n \leq c_3 \cdot \frac{\kappa \sqrt{\mu^+} \cdot w(\mathcal{T})}{\sqrt{n}} \,\bigg|\, \{\tilde{\boldsymbol{\eta}}_i\}\right) p(\tilde{\boldsymbol{\eta}}_1, \ldots, \tilde{\boldsymbol{\eta}}_n) \, d\tilde{\boldsymbol{\eta}}_1 \ldots d\tilde{\boldsymbol{\eta}}_n$$

$$\geq \int_{\mathcal{E}} \mathbb{P}_{\mathbf{X}}\left(\nu_n \leq c_3 \cdot \frac{\kappa \sqrt{\mu^+} \cdot w(\mathcal{T})}{\sqrt{n}} \,\bigg|\, \{\tilde{\boldsymbol{\eta}}_i\}\right) p(\tilde{\boldsymbol{\eta}}_1, \ldots, \tilde{\boldsymbol{\eta}}_n) \, d\tilde{\boldsymbol{\eta}}_1 \ldots d\tilde{\boldsymbol{\eta}}_n$$

$$\geq \left(1 - c_4 \exp\left(-\frac{w^2(\mathcal{T})}{8}\right)\right) \cdot \mathbb{P}(\mathcal{E})$$

$$\geq \left(1 - c_4 \exp\left(-\frac{w^2(\mathcal{T})}{8}\right)\right)(1 - 2\exp(-c_1 m))$$

$$\geq 1 - 2\exp(-c_1 m) - c_4 \exp\left(-\frac{w^2(\mathcal{T})}{8}\right)$$

$$\geq 1 - c_5 \exp(-c_6 m),$$

where the last step follows from $w(\mathcal{T}) = w(\mathbb{S}^{m \times m-1} \times \mathcal{C}) = w(\mathbb{S}^{m \times m-1}) + w(\mathcal{C}) = \Theta(m) + w(\mathcal{C})$. Since the invertibility for $\mathcal{M}$ is implied by the conditions of Lemma S.2, we have that if $n \geq \max\{n_0, C_0 \tau^4 m\}$,

$$\beta_n \leq c_7 \cdot \frac{\kappa \sqrt{\mu^+}(m + w(\mathcal{C}))}{\sqrt{n}}$$

with probability at least $1 - \epsilon - c_5 \exp(-c_6 m)$. Finally we complete the proof by renaming the constants. ∎

The proof of Lemma S.6 suggests that $\beta_n$ for any singleton $\mathcal{K}$ satisfies

$$\beta_n(\mathcal{K}) \leq C_3' \cdot \frac{\kappa \sqrt{\mu^+} \cdot w(\mathcal{C})}{\sqrt{n}}, \tag{S.22}$$

with probability $1 - C_2' \exp\left(-C_1' m\right)$ if $n \geq C_0' \tau^4 m$. Combined with Lemma S.3 and S.4, this immediately implies the error of both ordinary and oracle constrained least squares

$$e_{\text{odn}} = \frac{C' \kappa \sqrt{\mu^+}}{\mu^- \sqrt{m}} \cdot \frac{w(\mathcal{C})}{\sqrt{n}} \tag{S.23}$$

$$e_{\text{orc}} = \frac{C' \kappa \sqrt{\mu^+}}{\mu^- \sqrt{\text{Tr}(\boldsymbol{\Sigma}_*^{-1})}} \cdot \frac{w(\mathcal{C})}{\sqrt{n}} \tag{S.24}$$

For the well-initialized AltMin, most of the analysis stays the same, with the exception being $\beta_n(\mathcal{M}(e_0))$. With a small value of $e_0$, the index set $\mathcal{M}(e_0)$ in the definition of $\beta_n(\mathcal{M}(e_0))$ will shrink, so that we are able to sharpen the upper bound of $\beta_n(\mathcal{M}(e_0))$. The following lemma bounds the $\beta_n(\mathcal{M}(e_0))$ at $e_0 = \sqrt{\frac{\sigma_*^-}{\mu^+}}$.

**Lemma S.7** *Suppose that the conditions of Lemma S.2 are satisfied with probability $1 - \epsilon$ when $n \geq n_0$. Under the assumptions (A1) and (A2), if $n \geq \max\left\{n_0, C_0 \cdot \max\{\tau^4, \kappa^4, 1\} \cdot \max\{w^2(\mathcal{C}), \frac{m^3}{w^2(\mathcal{C})}, m^2\}\right\}$, the type-II NDI strength for $\mathcal{M}(e_0)$ with $e_0 = \sqrt{\frac{\sigma_*^-}{\mu^+}}$ satisfies*

$$\beta_n\left(\mathcal{M}(e_0)\right) \leq C_3 \cdot \frac{\kappa \sqrt{\mu^+} \cdot w(\mathcal{S})}{\sqrt{n}} \tag{S.25}$$

*with probability at least $1 - \epsilon - C_2 \exp\left(-C_1 \cdot \min\left\{w^2(\mathcal{S}), m\right\}\right)$.*

*Proof:* Throughout the proof, $e_0$ is set as $\sqrt{\frac{\sigma_*^-}{\mu^+}}$, and we will use the shorthand notation $\beta_n$ and $\mathcal{M}$ for $\beta_n(\mathcal{M}(e_0))$ and $\mathcal{M}(e_0)$. First we introduce the following notations

$$\mathcal{S}' = e_0 \cdot \mathcal{S} = \{e_0 \mathbf{u} \mid \mathbf{u} \in \mathcal{S}\}$$

$$\boldsymbol{\Gamma_w} = \mathbb{E}\left[\mathbf{X} \mathbf{w} \mathbf{w}^T \mathbf{X}^T\right]$$

$$\boldsymbol{\Sigma_\theta} = \boldsymbol{\Sigma}_* + \boldsymbol{\Gamma_{\theta - \theta_*}}$$

$$\hat{\boldsymbol{\Gamma}}_\mathbf{w} = -\frac{1}{n} \sum_{i=1}^{n} \mathbf{X}_i \mathbf{w} \boldsymbol{\eta}_i^T - \frac{1}{n} \sum_{i=1}^{n} \boldsymbol{\eta}_i \mathbf{w}^T \mathbf{X}_i + \frac{1}{n} \sum_{i=1}^{n} \mathbf{X}_i \mathbf{w} \mathbf{w}^T \mathbf{X}_i^T$$

$$\hat{\boldsymbol{\Sigma}}_\theta = \frac{1}{n} \sum_{i=1}^{n} \boldsymbol{\eta}_i \boldsymbol{\eta}_i^T + \hat{\boldsymbol{\Gamma}}_{\theta - \theta_*} = \frac{1}{n} \sum_{i=1}^{n} \left(\mathbf{y}_i - \mathbf{X}_i \boldsymbol{\theta}\right)\left(\mathbf{y}_i - \mathbf{X}_i \boldsymbol{\theta}\right)^T$$

Note that $\mu^- \leq \lambda_{\min}(\boldsymbol{\Gamma_w}) \leq \lambda_{\max}(\boldsymbol{\Gamma_w}) \leq \mu^+$ for any $\mathbf{w} \in \mathbb{S}^{p-1}$, $\boldsymbol{\Gamma_w} = \mathbb{E}[\hat{\boldsymbol{\Gamma}}_\mathbf{w}]$, $\boldsymbol{\Sigma_\theta} = \mathbb{E}[\hat{\boldsymbol{\Sigma}}_\theta]$ and $\mathcal{M} \subseteq \{\hat{\boldsymbol{\Sigma}}_\theta \mid \boldsymbol{\theta} \in \mathcal{S}' + \boldsymbol{\theta}_*\}$. Then we decompose $\beta_n$ as

$$\beta_n = \sup_{\boldsymbol{\Sigma} \in \mathcal{M}} \sup_{\mathbf{u} \in \mathcal{C}} \frac{2}{n} \sum_{i=1}^{n} \frac{\boldsymbol{\eta}_i^T \boldsymbol{\Sigma}^{-1} \mathbf{X}_i \mathbf{u}}{\|\boldsymbol{\Sigma}_*^{1/2} \boldsymbol{\Sigma}^{-1}\|_F} = \sup_{\boldsymbol{\Sigma} \in \mathcal{M}} \sup_{\mathbf{u} \in \mathcal{C}} \frac{2}{n} \sum_{i=1}^{n} \frac{\tilde{\boldsymbol{\eta}}_i^T \boldsymbol{\Sigma}_*^{1/2} \boldsymbol{\Sigma}^{-1} \mathbf{X}_i \mathbf{u}}{\|\boldsymbol{\Sigma}_*^{1/2} \boldsymbol{\Sigma}^{-1}\|_F}$$

$$\leq \sup_{\boldsymbol{\theta} \in \mathcal{S}' + \boldsymbol{\theta}_*} \sup_{\mathbf{u} \in \mathcal{C}} \frac{2}{n} \sum_{i=1}^{n} \tilde{\boldsymbol{\eta}}_i^T \left(\frac{\boldsymbol{\Sigma}_*^{1/2} \hat{\boldsymbol{\Sigma}}_\theta^{-1}}{\|\boldsymbol{\Sigma}_*^{1/2} \hat{\boldsymbol{\Sigma}}_\theta^{-1}\|_F} - \frac{\boldsymbol{\Sigma}_*^{1/2} \boldsymbol{\Sigma}_\theta^{-1}}{\|\boldsymbol{\Sigma}_*^{1/2} \boldsymbol{\Sigma}_\theta^{-1}\|_F}\right) \mathbf{X}_i \mathbf{u}$$

$$+ \sup_{\boldsymbol{\theta} \in \mathcal{S}' + \boldsymbol{\theta}_*} \sup_{\mathbf{u} \in \mathcal{C}} \frac{2}{n} \sum_{i=1}^{n} \frac{\tilde{\boldsymbol{\eta}}_i^T \boldsymbol{\Sigma}_*^{1/2} \boldsymbol{\Sigma}_\theta^{-1} \mathbf{X}_i \mathbf{u}}{\|\boldsymbol{\Sigma}_*^{1/2} \boldsymbol{\Sigma}_\theta^{-1}\|_F}$$

$$\leq \underbrace{\sup_{\boldsymbol{\Lambda} \in \mathbb{S}^{m \times m - 1}} \sup_{\mathbf{u} \in \mathcal{C}} \frac{2}{n} \sum_{i=1}^{n} \tilde{\boldsymbol{\eta}}_i^T \boldsymbol{\Lambda} \mathbf{X}_i \mathbf{u}}_{\nu_n} \cdot \underbrace{\sup_{\boldsymbol{\theta} \in \mathcal{S}' + \boldsymbol{\theta}_*} \left\|\frac{\boldsymbol{\Sigma}_*^{1/2} \hat{\boldsymbol{\Sigma}}_\theta^{-1}}{\|\boldsymbol{\Sigma}_*^{1/2} \hat{\boldsymbol{\Sigma}}_\theta^{-1}\|_F} - \frac{\boldsymbol{\Sigma}_*^{1/2} \boldsymbol{\Sigma}_\theta^{-1}}{\|\boldsymbol{\Sigma}_*^{1/2} \boldsymbol{\Sigma}_\theta^{-1}\|_F}\right\|_F}_{\zeta_n}$$

$$+ \underbrace{\sup_{\boldsymbol{\theta} \in \mathcal{S}' + \boldsymbol{\theta}_*} \sup_{\mathbf{u} \in \mathcal{C}} \frac{2}{n} \sum_{i=1}^{n} \frac{\tilde{\boldsymbol{\eta}}_i^T \boldsymbol{\Sigma}_*^{1/2} \boldsymbol{\Sigma}_\theta^{-1} \mathbf{X}_i \mathbf{u}}{\|\boldsymbol{\Sigma}_*^{1/2} \boldsymbol{\Sigma}_\theta^{-1}\|_F}}_{\phi_n}$$

where $\nu_n$ is analyzed in the proof of Lemma S.6. Therefore we focus on bounding $\zeta_n$ and $\phi_n$. We first try to bound $\zeta_n$,

$$
\begin{aligned}
\zeta_n &= \sup_{\boldsymbol{\theta}\in\mathcal{S}'+\boldsymbol{\theta}_*} \left\| \frac{\boldsymbol{\Sigma}_*^{1/2}\hat{\boldsymbol{\Sigma}}_{\boldsymbol{\theta}}^{-1}}{\|\boldsymbol{\Sigma}_*^{1/2}\hat{\boldsymbol{\Sigma}}_{\boldsymbol{\theta}}^{-1}\|_F} - \frac{\boldsymbol{\Sigma}_*^{1/2}\boldsymbol{\Sigma}_{\boldsymbol{\theta}}^{-1}}{\|\boldsymbol{\Sigma}_*^{1/2}\boldsymbol{\Sigma}_{\boldsymbol{\theta}}^{-1}\|_F} \right\|_F \\
&\leq \sup_{\boldsymbol{\theta}\in\mathcal{S}'+\boldsymbol{\theta}_*} \left\| \frac{\boldsymbol{\Sigma}_*^{1/2}\hat{\boldsymbol{\Sigma}}_{\boldsymbol{\theta}}^{-1}}{\|\boldsymbol{\Sigma}_*^{1/2}\hat{\boldsymbol{\Sigma}}_{\boldsymbol{\theta}}^{-1}\|_F} - \frac{\boldsymbol{\Sigma}_*^{1/2}\hat{\boldsymbol{\Sigma}}_{\boldsymbol{\theta}}^{-1}}{\|\boldsymbol{\Sigma}_*^{1/2}\boldsymbol{\Sigma}_{\boldsymbol{\theta}}^{-1}\|_F} \right\|_F + \sup_{\boldsymbol{\theta}\in\mathcal{S}'+\boldsymbol{\theta}_*} \left\| \frac{\boldsymbol{\Sigma}_*^{1/2}\hat{\boldsymbol{\Sigma}}_{\boldsymbol{\theta}}^{-1}}{\|\boldsymbol{\Sigma}_*^{1/2}\boldsymbol{\Sigma}_{\boldsymbol{\theta}}^{-1}\|_F} - \frac{\boldsymbol{\Sigma}_*^{1/2}\boldsymbol{\Sigma}_{\boldsymbol{\theta}}^{-1}}{\|\boldsymbol{\Sigma}_*^{1/2}\boldsymbol{\Sigma}_{\boldsymbol{\theta}}^{-1}\|_F} \right\|_F \\
&\leq \sup_{\boldsymbol{\theta}\in\mathcal{S}'+\boldsymbol{\theta}_*} \left| \frac{\|\boldsymbol{\Sigma}_*^{1/2}\hat{\boldsymbol{\Sigma}}_{\boldsymbol{\theta}}^{-1}\|_F - \|\boldsymbol{\Sigma}_*^{1/2}\boldsymbol{\Sigma}_{\boldsymbol{\theta}}^{-1}\|_F}{\|\boldsymbol{\Sigma}_*^{1/2}\boldsymbol{\Sigma}_{\boldsymbol{\theta}}^{-1}\|_F} \right| + \sup_{\boldsymbol{\theta}\in\mathcal{S}'+\boldsymbol{\theta}_*} \frac{\left\|\boldsymbol{\Sigma}_*^{1/2}\hat{\boldsymbol{\Sigma}}_{\boldsymbol{\theta}}^{-1} - \boldsymbol{\Sigma}_*^{1/2}\boldsymbol{\Sigma}_{\boldsymbol{\theta}}^{-1}\right\|_F}{\|\boldsymbol{\Sigma}_*^{1/2}\boldsymbol{\Sigma}_{\boldsymbol{\theta}}^{-1}\|_F} \\
&\leq 2 \sup_{\boldsymbol{\theta}\in\mathcal{S}'+\boldsymbol{\theta}_*} \frac{\left\|\boldsymbol{\Sigma}_*^{1/2}\hat{\boldsymbol{\Sigma}}_{\boldsymbol{\theta}}^{-1} - \boldsymbol{\Sigma}_*^{1/2}\boldsymbol{\Sigma}_{\boldsymbol{\theta}}^{-1}\right\|_F}{\|\boldsymbol{\Sigma}_*^{1/2}\boldsymbol{\Sigma}_{\boldsymbol{\theta}}^{-1}\|_F} \leq 2 \sup_{\boldsymbol{\theta}\in\mathcal{S}'+\boldsymbol{\theta}_*} \frac{\left\|\boldsymbol{\Sigma}_*^{1/2}(\hat{\boldsymbol{\Sigma}}_{\boldsymbol{\theta}}^{-1} - \boldsymbol{\Sigma}_{\boldsymbol{\theta}}^{-1})\boldsymbol{\Sigma}_*^{1/2}\right\|_2 \cdot \left\|\boldsymbol{\Sigma}_*^{-1/2}\right\|_F}{\lambda_{\min}\left(\boldsymbol{\Sigma}_*^{1/2}\boldsymbol{\Sigma}_{\boldsymbol{\theta}}^{-1}\boldsymbol{\Sigma}_*^{1/2}\right) \cdot \left\|\boldsymbol{\Sigma}_*^{-1/2}\right\|_F} \\
&\leq 2 \sup_{\boldsymbol{\theta}\in\mathcal{S}'+\boldsymbol{\theta}_*} \frac{\left\|\boldsymbol{\Sigma}_*^{-1/2}(\hat{\boldsymbol{\Sigma}}_{\boldsymbol{\theta}} - \boldsymbol{\Sigma}_{\boldsymbol{\theta}})\boldsymbol{\Sigma}_*^{-1/2}\right\|_2 \cdot \left\|\boldsymbol{\Sigma}_*^{1/2}\hat{\boldsymbol{\Sigma}}_{\boldsymbol{\theta}}^{-1}\boldsymbol{\Sigma}_*^{1/2}\right\|_2 \cdot \left\|\boldsymbol{\Sigma}_*^{1/2}\boldsymbol{\Sigma}_{\boldsymbol{\theta}}^{-1}\boldsymbol{\Sigma}_*^{1/2}\right\|_2}{\lambda_{\min}\left(\boldsymbol{\Sigma}_*^{1/2}\boldsymbol{\Sigma}_{\boldsymbol{\theta}}^{-1}\boldsymbol{\Sigma}_*^{1/2}\right)} \\
&= 2 \sup_{\boldsymbol{\theta}\in\mathcal{S}'+\boldsymbol{\theta}_*} \frac{\left\|\boldsymbol{\Sigma}_*^{-1/2}(\hat{\boldsymbol{\Sigma}}_{\boldsymbol{\theta}} - \boldsymbol{\Sigma}_{\boldsymbol{\theta}})\boldsymbol{\Sigma}_*^{-1/2}\right\|_2 \cdot \lambda_{\max}\left(\boldsymbol{\Sigma}_*^{-1/2}\boldsymbol{\Sigma}_{\boldsymbol{\theta}}\boldsymbol{\Sigma}_*^{-1/2}\right)}{\lambda_{\min}\left(\boldsymbol{\Sigma}_*^{-1/2}\hat{\boldsymbol{\Sigma}}_{\boldsymbol{\theta}}\boldsymbol{\Sigma}_*^{-1/2}\right) \cdot \lambda_{\min}\left(\boldsymbol{\Sigma}_*^{-1/2}\boldsymbol{\Sigma}_{\boldsymbol{\theta}}\boldsymbol{\Sigma}_*^{-1/2}\right)} \\
&\leq \frac{2\sup_{\boldsymbol{\theta}\in\mathcal{S}'+\boldsymbol{\theta}_*} \left\|\boldsymbol{\Sigma}_*^{-1/2}(\hat{\boldsymbol{\Sigma}}_{\boldsymbol{\theta}} - \boldsymbol{\Sigma}_{\boldsymbol{\theta}})\boldsymbol{\Sigma}_*^{-1/2}\right\|_2 \cdot \sup_{\mathbf{w}\in\mathcal{S}'} \lambda_{\max}\left(\boldsymbol{\Sigma}_*^{-1/2}(\boldsymbol{\Sigma}_* + \boldsymbol{\Gamma}_{\mathbf{w}})\boldsymbol{\Sigma}_*^{-1/2}\right)}{\inf_{\boldsymbol{\theta}\in\mathcal{S}'+\boldsymbol{\theta}_*} \lambda_{\min}\left(\boldsymbol{\Sigma}_*^{-1/2}\hat{\boldsymbol{\Sigma}}_{\boldsymbol{\theta}}\boldsymbol{\Sigma}_*^{-1/2}\right) \cdot \inf_{\mathbf{w}\in\mathcal{S}'} \lambda_{\min}\left(\boldsymbol{\Sigma}_*^{-1/2}(\boldsymbol{\Sigma}_* + \boldsymbol{\Gamma}_{\mathbf{w}})\boldsymbol{\Sigma}_*^{-1/2}\right)} \\
&\leq \frac{2\sup_{\boldsymbol{\theta}\in\mathcal{S}'+\boldsymbol{\theta}_*} \left\|\boldsymbol{\Sigma}_*^{-1/2}(\hat{\boldsymbol{\Sigma}}_{\boldsymbol{\theta}} - \boldsymbol{\Sigma}_{\boldsymbol{\theta}})\boldsymbol{\Sigma}_*^{-1/2}\right\|_2 \cdot \left(1 + \frac{\mu^+}{\sigma_*^-} \cdot \sup_{\mathbf{w}\in\mathcal{S}'} \|\mathbf{w}\|_2^2\right)}{(1 - 2\delta_n) \cdot \left(1 + \frac{\mu^-}{\sigma_*^+} \cdot \inf_{\mathbf{w}\in\mathcal{S}'} \|\mathbf{w}\|_2^2\right)} \\
&\leq 8 \sup_{\boldsymbol{\theta}\in\mathcal{S}'+\boldsymbol{\theta}_*} \left\|\boldsymbol{\Sigma}_*^{-1/2}(\hat{\boldsymbol{\Sigma}}_{\boldsymbol{\theta}} - \boldsymbol{\Sigma}_{\boldsymbol{\theta}})\boldsymbol{\Sigma}_*^{-1/2}\right\|_2
\end{aligned}
$$

where the last two steps use the conditions in Lemma S.2 and borrow some derivations from its proof. The last term can be further bounded as follows,

$$
\begin{aligned}
\sup_{\boldsymbol{\theta}\in\mathcal{S}'+\boldsymbol{\theta}_*} \left\|\boldsymbol{\Sigma}_*^{-\frac{1}{2}}(\hat{\boldsymbol{\Sigma}}_{\boldsymbol{\theta}} - \boldsymbol{\Sigma}_{\boldsymbol{\theta}})\boldsymbol{\Sigma}_*^{-\frac{1}{2}}\right\|_2 &= \sup_{\mathbf{w}\in\mathcal{S}'} \left\|\boldsymbol{\Sigma}_*^{-\frac{1}{2}}\left(\frac{1}{n}\sum_{i=1}^{n}\boldsymbol{\eta}_i\boldsymbol{\eta}_i^T + \hat{\boldsymbol{\Gamma}}_{\mathbf{w}} - \boldsymbol{\Sigma}_* - \boldsymbol{\Gamma}_{\mathbf{w}}\right)\boldsymbol{\Sigma}_*^{-\frac{1}{2}}\right\|_2 \\
&\leq \left\|\frac{1}{n}\sum_{i=1}^{n}\tilde{\boldsymbol{\eta}}_i\tilde{\boldsymbol{\eta}}_i^T - \mathbf{I}\right\|_2 + \sup_{\mathbf{w}\in\mathcal{S}'}\left(\left\|\frac{1}{n}\sum_{i=1}^{n}\boldsymbol{\Sigma}_*^{-\frac{1}{2}}\mathbf{X}_i\mathbf{w}\tilde{\boldsymbol{\eta}}_i^T\right\|_2 + \left\|\frac{1}{n}\sum_{i=1}^{n}\tilde{\boldsymbol{\eta}}_i\mathbf{w}^T\mathbf{X}_i^T\boldsymbol{\Sigma}_*^{-\frac{1}{2}}\right\|_2\right) \\
&\quad + \sup_{\mathbf{w}\in\mathcal{S}'}\left\|\boldsymbol{\Sigma}_*^{-1/2}\left(\frac{1}{n}\sum_{i=1}^{n}\mathbf{X}_i\mathbf{w}\mathbf{w}^T\mathbf{X}_i^T - \boldsymbol{\Gamma}_{\mathbf{w}}\right)\boldsymbol{\Sigma}_*^{-1/2}\right\|_2 \\
&\leq \delta_n + \frac{e_0}{\sqrt{\sigma_*^-}} \cdot \sup_{\mathbf{w}\in\mathcal{C}}\left\|\frac{2}{n}\sum_{i=1}^{n}\mathbf{X}_i\mathbf{w}\tilde{\boldsymbol{\eta}}_i^T\right\|_2 + \frac{e_0^2}{\sigma_*^-} \cdot \sup_{\mathbf{w}\in\mathcal{C}}\left\|\frac{1}{n}\sum_{i=1}^{n}\mathbf{X}_i\mathbf{w}\mathbf{w}^T\mathbf{X}_i^T - \boldsymbol{\Gamma}_{\mathbf{w}}\right\|_2 \\
&\leq \delta_n + \frac{e_0\gamma_n}{\sqrt{\sigma_*^-}} + \frac{e_0^2}{\sigma_*^-} \cdot \sup_{\mathbf{v}\in\mathbb{S}^{m-1}}\sup_{\mathbf{w}\in\mathcal{C}}\left|\frac{1}{n}\sum_{i=1}^{n}(\mathbf{w}^T\mathbf{X}_i^T\mathbf{v})^2 - \mathbb{E}(\mathbf{w}^T\mathbf{X}^T\mathbf{v})^2\right| \\
&\leq c_1\tau^2\sqrt{\frac{m}{n}} + \frac{c_2\kappa\tau(\sqrt{m} + w(\mathcal{C}))}{\sqrt{n}} + \frac{c_3\kappa^2(\sqrt{m} + w(\mathcal{C}))}{\sqrt{n}}
\end{aligned}
$$

which holds with probability at least $1 - c_4\exp(-c_5 m)$ when $n \geq c_6 \max\left\{\tau^4, 1\right\} \cdot \max\left\{w^2(\mathcal{C}), m\right\}$. The last step follows from Proposition S.2, Lemma S.5 and intermediate results in the proof of Lemma

S.4. Hence $\zeta_n$ can be bounded by

$$\zeta_n \le c_7 \cdot \max\left\{\tau^2, \kappa^2\right\} \cdot \frac{\sqrt{m} + w(\mathcal{C})}{\sqrt{n}}$$

Now we turn to bounding $\phi_n$. Following the idea for proving Lemma S.6, we also consider the randomness of $\{\tilde{\boldsymbol{\eta}}_i\}$ and $\{\mathbf{X}_i\}$ sequentially. For $\{\tilde{\boldsymbol{\eta}}_i\}$, we first have that the event

$$\mathcal{E} = \left\{\{\tilde{\boldsymbol{\eta}}_i\} \ \bigg| \ \sup_{\boldsymbol{\Lambda} \in \mathbb{S}^{m \times m-1}} \frac{1}{n} \sum_{i=1}^{n} \left\|\boldsymbol{\Lambda}^T \tilde{\boldsymbol{\eta}}_i\right\|_2^2 \le 2\right\}$$

holds with probability at least $1 - 2\exp(-c_1' m)$ if $n \ge c_0' \tau^4 m$, which is shown in the proof of Lemma S.6. Now we consider the randomness of $\{\mathbf{X}_i\}$ under any fixed $\{\boldsymbol{\eta}_i\} \in \mathcal{E}$. We have

$$\phi_n = \sup_{\boldsymbol{\theta} \in \mathcal{S}' + \boldsymbol{\theta}_*} \sup_{\mathbf{u} \in \mathcal{C}} \frac{2}{n} \sum_{i=1}^{n} \frac{\tilde{\boldsymbol{\eta}}_i^T \boldsymbol{\Sigma}_*^{1/2} \boldsymbol{\Sigma}_{\boldsymbol{\theta}}^{-1} \mathbf{X}_i \mathbf{u}}{\|\boldsymbol{\Sigma}_*^{1/2} \boldsymbol{\Sigma}_{\boldsymbol{\theta}}^{-1}\|_F}$$

$$\le \frac{1}{e_0} \cdot \sup_{\mathbf{w} \in \mathcal{S}'} \sup_{\mathbf{u} \in \mathcal{S}'} \frac{2}{n} \sum_{i=1}^{n} \frac{\tilde{\boldsymbol{\eta}}_i^T \boldsymbol{\Sigma}_*^{1/2} (\boldsymbol{\Sigma}_* + \boldsymbol{\Gamma}_\mathbf{w})^{-1} \mathbf{X}_i \mathbf{u}}{\|\boldsymbol{\Sigma}_*^{1/2} (\boldsymbol{\Sigma}_* + \boldsymbol{\Gamma}_\mathbf{w})^{-1}\|_F}$$

$$= \frac{2}{e_0 \sqrt{n}} \cdot \sup_{\mathbf{t} \in \mathcal{T}} Z_\mathbf{t} \ ,$$

where $Z_\mathbf{t} = \frac{1}{\sqrt{n}} \sum_{i=1}^{n} \frac{\tilde{\boldsymbol{\eta}}_i^T \boldsymbol{\Sigma}_*^{1/2} (\boldsymbol{\Sigma}_* + \boldsymbol{\Gamma}_\mathbf{w})^{-1} \mathbf{X}_i \mathbf{u}}{\|\boldsymbol{\Sigma}_*^{1/2} (\boldsymbol{\Sigma}_* + \boldsymbol{\Gamma}_\mathbf{w})^{-1}\|_F}$, $\mathbf{t} = (\mathbf{w}, \mathbf{u})$ and $\mathcal{T} = \mathcal{S}' \times \mathcal{S}'$. Note that

$$\forall \, \mathbf{t}, \mathbf{t}' \in \mathcal{T}, \quad \|\mathbf{t} - \mathbf{t}'\|_2 = \sqrt{\|\mathbf{w} - \mathbf{w}'\|_F^2 + \|\mathbf{u} - \mathbf{u}'\|_2^2} \le 2\sqrt{2}e_0 \quad \implies \quad \operatorname{diam}(\mathcal{T}) \le 2\sqrt{2}e_0$$

Then we try to bound the stochastic process $\{Z_\mathbf{t}\}_{\mathbf{t} \in \mathcal{T}}$ using Theorem S.1. We start with verifying the required condition.

$$\forall \, \mathbf{t}, \mathbf{t}' \in \mathcal{T},$$

$$\|Z_\mathbf{t} - Z_{\mathbf{t}'}\|_{\psi_2}$$

$$= \left\|\frac{1}{\sqrt{n}} \sum_{i=1}^{n} \frac{\tilde{\boldsymbol{\eta}}_i^T \boldsymbol{\Sigma}_*^{1/2} (\boldsymbol{\Sigma}_* + \boldsymbol{\Gamma}_\mathbf{w})^{-1} \mathbf{X}_i \mathbf{u}}{\|\boldsymbol{\Sigma}_*^{1/2} (\boldsymbol{\Sigma}_* + \boldsymbol{\Gamma}_\mathbf{w})^{-1}\|_F} - \frac{1}{\sqrt{n}} \sum_{i=1}^{n} \frac{\tilde{\boldsymbol{\eta}}_i^T \boldsymbol{\Sigma}_*^{1/2} (\boldsymbol{\Sigma}_* + \boldsymbol{\Gamma}_{\mathbf{w}'})^{-1} \mathbf{X}_i \mathbf{u}'}{\|\boldsymbol{\Sigma}_*^{1/2} (\boldsymbol{\Sigma}_* + \boldsymbol{\Gamma}_{\mathbf{w}'})^{-1}\|_F}\right\|_{\psi_2}$$

$$\le \left\|\frac{1}{\sqrt{n}} \sum_{i=1}^{n} \tilde{\boldsymbol{\eta}}_i^T \left(\frac{\boldsymbol{\Sigma}_*^{1/2} (\boldsymbol{\Sigma}_* + \boldsymbol{\Gamma}_\mathbf{w})^{-1}}{\|\boldsymbol{\Sigma}_*^{1/2} (\boldsymbol{\Sigma}_* + \boldsymbol{\Gamma}_\mathbf{w})^{-1}\|_F} - \frac{\boldsymbol{\Sigma}_*^{1/2} (\boldsymbol{\Sigma}_* + \boldsymbol{\Gamma}_{\mathbf{w}'})^{-1}}{\|\boldsymbol{\Sigma}_*^{1/2} (\boldsymbol{\Sigma}_* + \boldsymbol{\Gamma}_{\mathbf{w}'})^{-1}\|_F}\right) \mathbf{X}_i \mathbf{u}\right\|_{\psi_2}$$

$$+ \left\|\frac{1}{\sqrt{n}} \sum_{i=1}^{n} \frac{\tilde{\boldsymbol{\eta}}_i^T \boldsymbol{\Sigma}_*^{1/2} (\boldsymbol{\Sigma}_* + \boldsymbol{\Gamma}_{\mathbf{w}'})^{-1} \mathbf{X}_i (\mathbf{u} - \mathbf{u}')}{\|\boldsymbol{\Sigma}_*^{1/2} (\boldsymbol{\Sigma}_* + \boldsymbol{\Gamma}_{\mathbf{w}'})^{-1}\|_F}\right\|_{\psi_2}$$

$$\overset{(a)}{\le} c_2' \sqrt{\frac{1}{n} \sum_{i=1}^{n} \left\|\left(\frac{\boldsymbol{\Sigma}_*^{\frac{1}{2}} (\boldsymbol{\Sigma}_* + \boldsymbol{\Gamma}_\mathbf{w})^{-1}}{\|\boldsymbol{\Sigma}_*^{\frac{1}{2}} (\boldsymbol{\Sigma}_* + \boldsymbol{\Gamma}_\mathbf{w})^{-1}\|_F} - \frac{\boldsymbol{\Sigma}_*^{\frac{1}{2}} (\boldsymbol{\Sigma}_* + \boldsymbol{\Gamma}_{\mathbf{w}'})^{-1}}{\|\boldsymbol{\Sigma}_*^{\frac{1}{2}} (\boldsymbol{\Sigma}_* + \boldsymbol{\Gamma}_{\mathbf{w}'})^{-1}\|_F}\right)^T \tilde{\boldsymbol{\eta}}_i\right\|_2^2} \cdot \sup_{\mathbf{v} \in \mathbb{S}^{m-1}} \left\|\mathbf{v}^T \mathbf{X} \mathbf{u}\right\|_{\psi_2}$$

$$+ c_2' \sqrt{\frac{1}{n} \sum_{i=1}^{n} \left\|\left(\frac{\boldsymbol{\Sigma}_*^{\frac{1}{2}} (\boldsymbol{\Sigma}_* + \boldsymbol{\Gamma}_{\mathbf{w}'})^{-1}}{\|\boldsymbol{\Sigma}_*^{\frac{1}{2}} (\boldsymbol{\Sigma}_* + \boldsymbol{\Gamma}_{\mathbf{w}'})^{-1}\|_F}\right)^T \tilde{\boldsymbol{\eta}}_i\right\|_2^2} \cdot \|\mathbf{u} - \mathbf{u}'\|_2 \cdot \sup_{\mathbf{v} \in \mathbb{S}^{m-1}} \left\|\mathbf{v}^T \mathbf{X} \frac{\mathbf{u} - \mathbf{u}'}{\|\mathbf{u} - \mathbf{u}'\|_2}\right\|_{\psi_2}$$

$$\overset{(b)}{\le} \sqrt{2} c_2' \kappa \sqrt{\mu^+} \left(e_0 \left\|\frac{\boldsymbol{\Sigma}_*^{\frac{1}{2}} (\boldsymbol{\Sigma}_* + \boldsymbol{\Gamma}_\mathbf{w})^{-1}}{\|\boldsymbol{\Sigma}_*^{\frac{1}{2}} (\boldsymbol{\Sigma}_* + \boldsymbol{\Gamma}_\mathbf{w})^{-1}\|_F} - \frac{\boldsymbol{\Sigma}_*^{\frac{1}{2}} (\boldsymbol{\Sigma}_* + \boldsymbol{\Gamma}_{\mathbf{w}'})^{-1}}{\|\boldsymbol{\Sigma}_*^{\frac{1}{2}} (\boldsymbol{\Sigma}_* + \boldsymbol{\Gamma}_{\mathbf{w}'})^{-1}\|_F}\right\|_F + \|\mathbf{u} - \mathbf{u}'\|_2\right)$$

$$\overset{(c)}{\le} \sqrt{2} c_2' \kappa \sqrt{\mu^+} \left(8 \|\mathbf{w} - \mathbf{w}'\|_2 + \|\mathbf{u} - \mathbf{u}'\|_2\right)$$

$$\le 16 c_2' \kappa \sqrt{\mu^+} \left\|\begin{bmatrix} \mathbf{w} \\ \mathbf{u} \end{bmatrix} - \begin{bmatrix} \mathbf{w}' \\ \mathbf{u}' \end{bmatrix}\right\|_2 \quad \implies \quad K = 16 c_2' \kappa \sqrt{\mu^+} \ ,$$

where step (a) follows from Lemma S.1 and step (b) follows from the event $\mathcal{E}$. Step (c) follows from the calculation below (similar to bounding $\zeta_n$),

$$
\left\| \frac{\boldsymbol{\Sigma}_*^{1/2}(\boldsymbol{\Sigma}_* + \boldsymbol{\Gamma}_{\mathbf{w}})^{-1}}{\|\boldsymbol{\Sigma}_*^{1/2}(\boldsymbol{\Sigma}_* + \boldsymbol{\Gamma}_{\mathbf{w}})^{-1}\|_F} - \frac{\boldsymbol{\Sigma}_*^{1/2}(\boldsymbol{\Sigma}_* + \boldsymbol{\Gamma}_{\mathbf{w}'})^{-1}}{\|\boldsymbol{\Sigma}_*^{1/2}(\boldsymbol{\Sigma}_* + \boldsymbol{\Gamma}_{\mathbf{w}'})^{-1}\|_F} \right\|_F
$$

$$
\leq \left\| \frac{\boldsymbol{\Sigma}_*^{1/2}(\boldsymbol{\Sigma}_* + \boldsymbol{\Gamma}_{\mathbf{w}})^{-1}}{\|\boldsymbol{\Sigma}_*^{1/2}(\boldsymbol{\Sigma}_* + \boldsymbol{\Gamma}_{\mathbf{w}})^{-1}\|_F} - \frac{\boldsymbol{\Sigma}_*^{1/2}(\boldsymbol{\Sigma}_* + \boldsymbol{\Gamma}_{\mathbf{w}'})^{-1}}{\|\boldsymbol{\Sigma}_*^{1/2}(\boldsymbol{\Sigma}_* + \boldsymbol{\Gamma}_{\mathbf{w}})^{-1}\|_F} \right\|_F
$$

$$
+ \left\| \frac{\boldsymbol{\Sigma}_*^{1/2}(\boldsymbol{\Sigma}_* + \boldsymbol{\Gamma}_{\mathbf{w}'})^{-1}}{\|\boldsymbol{\Sigma}_*^{1/2}(\boldsymbol{\Sigma}_* + \boldsymbol{\Gamma}_{\mathbf{w}})^{-1}\|_F} - \frac{\boldsymbol{\Sigma}_*^{1/2}(\boldsymbol{\Sigma}_* + \boldsymbol{\Gamma}_{\mathbf{w}'})^{-1}}{\|\boldsymbol{\Sigma}_*^{1/2}(\boldsymbol{\Sigma}_* + \boldsymbol{\Gamma}_{\mathbf{w}'})^{-1}\|_F} \right\|_F
$$

$$
\leq \frac{2 \left\| \boldsymbol{\Sigma}_*^{1/2}(\boldsymbol{\Sigma}_* + \boldsymbol{\Gamma}_{\mathbf{w}})^{-1} - \boldsymbol{\Sigma}_*^{1/2}(\boldsymbol{\Sigma}_* + \boldsymbol{\Gamma}_{\mathbf{w}'})^{-1} \right\|_F}{\left\| \boldsymbol{\Sigma}_*^{1/2}(\boldsymbol{\Sigma}_* + \boldsymbol{\Gamma}_{\mathbf{w}})^{-1} \right\|_F}
$$

$$
\leq \frac{2 \left\| \boldsymbol{\Sigma}_*^{1/2} \left( (\boldsymbol{\Sigma}_* + \boldsymbol{\Gamma}_{\mathbf{w}})^{-1} - (\boldsymbol{\Sigma}_* + \boldsymbol{\Gamma}_{\mathbf{w}'})^{-1} \right) \boldsymbol{\Sigma}_*^{1/2} \right\|_2}{\lambda_{\min} \left( \boldsymbol{\Sigma}_*^{1/2}(\boldsymbol{\Sigma}_* + \boldsymbol{\Gamma}_{\mathbf{w}})^{-1} \boldsymbol{\Sigma}_*^{1/2} \right)}
$$

$$
\leq \frac{2 \left\| \boldsymbol{\Sigma}_*^{-1/2} \left( \boldsymbol{\Gamma}_{\mathbf{w}} - \boldsymbol{\Gamma}_{\mathbf{w}'} \right) \boldsymbol{\Sigma}_*^{-1/2} \right\|_2 \cdot \left\| \boldsymbol{\Sigma}_*^{1/2}(\boldsymbol{\Sigma}_* + \boldsymbol{\Gamma}_{\mathbf{w}})^{-1} \boldsymbol{\Sigma}_*^{1/2} \right\|_2 \cdot \left\| \boldsymbol{\Sigma}_*^{1/2}(\boldsymbol{\Sigma}_* + \boldsymbol{\Gamma}_{\mathbf{w}'})^{-1} \boldsymbol{\Sigma}_*^{1/2} \right\|_2}{\lambda_{\min} \left( \boldsymbol{\Sigma}_*^{1/2}(\boldsymbol{\Sigma}_* + \boldsymbol{\Gamma}_{\mathbf{w}})^{-1} \boldsymbol{\Sigma}_*^{1/2} \right)}
$$

$$
= \frac{2 \left\| \boldsymbol{\Sigma}_*^{-1/2} \left( \boldsymbol{\Gamma}_{\mathbf{w}} - \boldsymbol{\Gamma}_{\mathbf{w}'} \right) \boldsymbol{\Sigma}_*^{-1/2} \right\|_2 \cdot \lambda_{\max} \left( \boldsymbol{\Sigma}_*^{-1/2}(\boldsymbol{\Sigma}_* + \boldsymbol{\Gamma}_{\mathbf{w}}) \boldsymbol{\Sigma}_*^{-1/2} \right)}{\lambda_{\min} \left( \boldsymbol{\Sigma}_*^{-1/2}(\boldsymbol{\Sigma}_* + \boldsymbol{\Gamma}_{\mathbf{w}'}) \boldsymbol{\Sigma}_*^{-1/2} \right) \cdot \lambda_{\min} \left( \boldsymbol{\Sigma}_*^{-1/2}(\boldsymbol{\Sigma}_* + \boldsymbol{\Gamma}_{\mathbf{w}}) \boldsymbol{\Sigma}_*^{-1/2} \right)}
$$

$$
\leq \frac{2 \|\boldsymbol{\Gamma}_{\mathbf{w}} - \boldsymbol{\Gamma}_{\mathbf{w}'}\|_2 \cdot \left( 1 + \frac{\mu^+}{\sigma_*^-} \|\mathbf{w}\|_2^2 \right)}{\sigma_*^- \left( 1 + \frac{\mu^-}{\sigma_*^+} \|\mathbf{w}'\|_2^2 \right) \cdot \left( 1 + \frac{\mu^-}{\sigma_*^+} \|\mathbf{w}\|_2^2 \right)} \leq \frac{4}{\sigma_*^-} \left\| \mathbb{E}\left[ \mathbf{X}\mathbf{w}\mathbf{w}^T\mathbf{X}^T \right] - \mathbb{E}\left[ \mathbf{X}\mathbf{w}'\mathbf{w}'^T\mathbf{X}^T \right] \right\|_2
$$

$$
\leq \frac{4}{\sigma_*^-} \cdot \sup_{\mathbf{v} \in \mathbb{S}^{m-1}} \left| \mathbf{v}^T \left( \mathbb{E}\left[ \mathbf{X}\mathbf{w}\mathbf{w}^T\mathbf{X}^T \right] - \mathbb{E}\left[ \mathbf{X}\mathbf{w}'\mathbf{w}'^T\mathbf{X}^T \right] \right) \mathbf{v} \right|
$$

$$
\leq \frac{4}{\sigma_*^-} \left( \sup_{\mathbf{v} \in \mathbb{S}^{m-1}} \left| \mathbf{v}^T \mathbb{E}\left[ \mathbf{X}\mathbf{w}(\mathbf{w} - \mathbf{w}')^T\mathbf{X}^T \right] \mathbf{v} \right| + \sup_{\mathbf{v} \in \mathbb{S}^{m-1}} \left| \mathbf{v}^T \mathbb{E}\left[ \mathbf{X}(\mathbf{w} - \mathbf{w}')\mathbf{w}'^T\mathbf{X}^T \right] \mathbf{v} \right| \right)
$$

$$
\leq \frac{8}{\sigma_*^-} \cdot \|\mathbf{w} - \mathbf{w}'\|_2 \cdot \sup_{\mathbf{v} \in \mathbb{S}^{m-1}} \sup_{\mathbf{z} \in \mathbb{S}^{p-1}} \sup_{\mathbf{r} \in \mathcal{S}'} \mathbf{v}^T \mathbb{E}\left[ \mathbf{X}\mathbf{r}\mathbf{z}^T\mathbf{X}^T \right] \mathbf{v}
$$

$$
\leq \frac{8e_0}{\sigma_*^-} \cdot \|\mathbf{w} - \mathbf{w}'\|_2 \cdot \sup_{\mathbf{v} \in \mathbb{S}^{m-1}} \sup_{\mathbf{z} \in \mathbb{S}^{p-1}} \sup_{\mathbf{r} \in \mathcal{S}'} \frac{\mathbb{E}\left( \frac{\mathbf{v}^T\mathbf{X}\mathbf{r}}{e_0} \right)^2 + \mathbb{E}\left( \mathbf{v}^T\mathbf{X}\mathbf{z} \right)^2}{2}
$$

$$
\leq \frac{8e_0}{\sigma_*^-} \cdot \|\mathbf{w} - \mathbf{w}'\|_2 \cdot \mu^+ = \frac{8}{e_0} \cdot \|\mathbf{w} - \mathbf{w}'\|_2
$$

By invoking Theorem S.1, we have for $\phi_n$ with any fixed $\{\tilde{\boldsymbol{\eta}}_i\} \in \mathcal{E}$,

$$
\phi_n = \frac{2}{e_0\sqrt{n}} \cdot \sup_{\mathbf{t} \in \mathcal{T}} Z_{\mathbf{t}} \leq \frac{2}{e_0\sqrt{n}} \cdot \sup_{\mathbf{t}, \mathbf{t}' \in \mathcal{T}} |Z_{\mathbf{t}} - Z_{\mathbf{t}'}| \leq \frac{2c_3'}{e_0} \cdot \frac{\kappa\sqrt{\mu^+} \cdot w(\mathcal{T})}{\sqrt{n}} = 4c_3' \cdot \frac{\kappa\sqrt{\mu^+} \cdot w(\mathcal{S})}{\sqrt{n}}
$$

with probability at least $1 - c_4' \exp\left( -\frac{w^2(\mathcal{T})}{\text{diam}^2(\mathcal{T})} \right) \geq 1 - c_4' \exp\left( -\frac{w^2(\mathcal{S})}{2} \right)$. Now we combine the randomness of $\mathbf{X}_i$ and $\tilde{\boldsymbol{\eta}}_i$, and get

$$
\mathbb{P}_{\mathbf{X}, \tilde{\boldsymbol{\eta}}} \left( \phi_n \leq 4c_3' \cdot \frac{\kappa\sqrt{\mu^+} \cdot w(\mathcal{S})}{\sqrt{n}} \right)
$$

$$
= \int \mathbb{P}_{\mathbf{X}} \left( \phi_n \leq 4c_3' \cdot \frac{\kappa\sqrt{\mu^+} \cdot w(\mathcal{S})}{\sqrt{n}} \,\middle|\, \{\tilde{\boldsymbol{\eta}}_i\} \right) p\left( \tilde{\boldsymbol{\eta}}_1, \ldots, \tilde{\boldsymbol{\eta}}_n \right) d\tilde{\boldsymbol{\eta}}_1 \ldots d\tilde{\boldsymbol{\eta}}_n
$$

$$\geq \int_{\mathcal{E}} \mathbb{P}_{\mathbf{X}} \left( \phi_n \leq 4c_3' \cdot \frac{\kappa \sqrt{\mu^+} \cdot w(\mathcal{S})}{\sqrt{n}} \ \bigg| \ \{\tilde{\boldsymbol{\eta}}_i\} \right) p\left(\tilde{\boldsymbol{\eta}}_1, \ldots, \tilde{\boldsymbol{\eta}}_n\right) d\tilde{\boldsymbol{\eta}}_1 \ldots d\tilde{\boldsymbol{\eta}}_n$$

$$\geq \left( 1 - c_4' \exp\left( -\frac{w^2(\mathcal{S})}{2} \right) \right) \cdot \mathbb{P}(\mathcal{E})$$

$$\geq \left( 1 - c_4' \exp\left( -\frac{w^2(\mathcal{S})}{2} \right) \right) (1 - 2\exp(-c_1' m))$$

$$\geq 1 - 2\exp(-c_1' m) - c_4' \exp\left( -\frac{w^2(\mathcal{S})}{2} \right)$$

We obtain the final bound by assembling everything above. If $n \geq \max\{n_0, C_0' \cdot \max\{\tau^4, 1\} \cdot \max\{w^2(\mathcal{C}), m\}\}$, with probability at least $1 - \epsilon - C_1' \exp\left(-C_2' \min\{w^2(\mathcal{S}), m\}\right)$, we have

$$\beta_n \leq \sqrt{m} \gamma_n \zeta_n + \phi_n$$

$$\leq C_3' \cdot \max\{\tau^2, \kappa^2\} \cdot \frac{\kappa\sqrt{\mu^+}(m + w(\mathcal{C}))(\sqrt{m} + w(\mathcal{C}))}{n} + C_4' \cdot \frac{\kappa\sqrt{\mu^+} \cdot w(\mathcal{S})}{\sqrt{n}} ,$$

In particular, if the sample size also satisfies $n \geq C_5' \cdot \max\left\{\tau^4, \kappa^4\right\} \cdot \max\left\{\frac{m^3}{w^2(\mathcal{C})}, m^2, w^2(\mathcal{C})\right\} \geq C_6' \cdot \max\left\{\tau^4, \kappa^4\right\} \cdot \left(\frac{(m+w(\mathcal{C}))(\sqrt{m}+w(\mathcal{C}))}{w(\mathcal{S})}\right)^2$, we further have

$$\beta_n \leq C_7' \cdot \frac{\kappa\sqrt{\mu^+} \cdot w(\mathcal{S})}{\sqrt{n}} ,$$

which completes the proof. ∎

# 4  Proofs of Lemma 1, 2 and Theorem 1, 2

Based on the analysis presented in Section 2 and 3, now we give the proof sketches of the main results shown in the paper.

**Statement of Lemma 1:**  *Under the assumptions (A1) and (A2), if the sample size $n \geq C_0 \max\left\{1, \tau^4, \kappa^4 \left(\frac{\sigma_*^+ \mu^+}{\sigma_*^- \mu^-}\right)^2\right\} \cdot \max\left\{m, \frac{w^4(\mathcal{C})}{m}\right\}$, with probability at least $1 - C_2 \exp(-C_1 m)$, $\hat{\boldsymbol{\Sigma}}(\boldsymbol{\theta})$ given in (13) is invertible for any $\boldsymbol{\theta} \in \mathcal{R}$ and its error satisfies*

$$d_1\left(\hat{\boldsymbol{\Sigma}}(\boldsymbol{\theta}), \ \boldsymbol{\Sigma}_*\right) \leq C_3 \tau^2 \sqrt{\frac{m}{n}} + C_4 \sqrt{\frac{\mu^+}{\sigma_*^-}} \cdot d_2\left(\boldsymbol{\theta}, \boldsymbol{\theta}_*\right) . \tag{S.26}$$

*Proof:*  The above error bound for $d_1$ directly follows from the deterministic bound in Lemma S.2 and the probabilistic bounds for $\alpha_n^-$, $\alpha_n^+$ (Lemma S.4), $\delta_n$ (Proposition S.2) and $\gamma_n$ (Lemma S.5). ∎

**Statement of Lemma 2:**  *Under the assumptions (A1) and (A2), if the sample size $n \geq C_0 \max\left\{1, \tau^4, \kappa^4 \left(\frac{\sigma_*^+ \mu^+}{\sigma_*^- \mu^-}\right)^2\right\} \cdot \max\left\{m, \frac{w^4(\mathcal{C})}{m}\right\}$, then with probability at least $1 - C_2 \exp(-C_1 m)$, the following bound holds for $\hat{\boldsymbol{\theta}}(\boldsymbol{\Sigma})$ given in (14) with any input $\boldsymbol{\Sigma} \in \mathcal{M}(+\infty)$,*

$$d_2\left(\hat{\boldsymbol{\theta}}(\boldsymbol{\Sigma}), \ \boldsymbol{\theta}_*\right) \leq (1 + d_1(\boldsymbol{\Sigma}, \boldsymbol{\Sigma}_*)) \cdot \frac{C_4 \kappa \sqrt{\mu^+}}{\mu^- \sqrt{\mathrm{Tr}(\boldsymbol{\Sigma}_*^{-1})}} \cdot \frac{m + w(\mathcal{C})}{\sqrt{n}} , \tag{S.27}$$

*where $\xi(\boldsymbol{\Sigma})$ is given in Definition 1.*

*Proof:* The above error bound for $d_2$ directly follows from the deterministic bound in Lemma S.3 and the probabilistic bounds for $\alpha_n^-, \alpha_n^+$ (Lemma S.4), $\delta_n$ (Proposition S.2), $\gamma_n$ (Lemma S.5) and $\beta_n(\mathcal{M}(e_0))$ with $e_0 = +\infty$ (Lemma S.6). ∎

**Statement of Theorem 1:** *Under the assumptions (A1) and (A2), if the sample size $n \geq C_0 \cdot \max\left\{1, \tau^4, \kappa^4 \left(\frac{\mu^+ \sigma_*^+}{\mu^- \sigma_*^-}\right)^2, \kappa^2 \left(\frac{\mu^+}{\mu^-}\right)^2 \left(\frac{\sigma_*^+}{\sigma_*^-}\right)\right\} \cdot \max\left\{\frac{w^4(\mathcal{C})}{m}, m\right\}$, and $\hat{\boldsymbol{\theta}}_{(0)}$ is a feasible initialization (i.e., $f(\hat{\boldsymbol{\theta}}_{(0)}) \leq f(\boldsymbol{\theta}_*)$), then with probability at least $1 - C_2 \exp(-C_1 m)$, the following error bound holds for $\hat{\boldsymbol{\theta}}_{(T)}$ returned by Algorithm 1*

$$\left\|\hat{\boldsymbol{\theta}}_{(T)} - \boldsymbol{\theta}_*\right\|_2 \leq e_{\min} + \rho_n^T \cdot \left(\left\|\hat{\boldsymbol{\theta}}_{(0)} - \boldsymbol{\theta}_*\right\|_2 - e_{\min}\right) , \tag{S.28}$$

*in which $\rho_n$ and $e_{\min}$ satisfy the inequalities below with $\delta_n = C_5 \tau^2 \sqrt{\frac{m}{n}} \leq \frac{1}{4}$,*

$$\rho_n \leq \frac{C_3 \kappa \mu^+}{\mu^- \sqrt{\sigma_*^- \operatorname{Tr}(\boldsymbol{\Sigma}_*^{-1})}} \cdot \frac{m + w(\mathcal{C})}{\sqrt{n}} \leq \frac{1}{2}, \quad e_{\min} \leq \frac{C_4 \kappa \sqrt{\mu^+}}{\mu^- \sqrt{\operatorname{Tr}(\boldsymbol{\Sigma}_*^{-1})}} \cdot \frac{m + w(\mathcal{C})}{\sqrt{n}} \cdot \frac{1 + \delta_n}{1 - \rho_n},$$
$$\tag{S.29}$$

*Proof:* The above error bound for AltMin directly follows from the deterministic bound in Theorem S.3 and the probabilistic bounds for $\alpha_n^-, \alpha_n^+$ (Lemma S.4), $\delta_n$ (Proposition S.2), $\gamma_n$ (Lemma S.5) and $\beta_n(\mathcal{M}(e_0))$ with $e_0 = +\infty$ (Lemma S.6). ∎

**Statement of Theorem 2:** *Under the assumptions (A1) and (A2), if the sample size $n \geq C_0 \cdot \max\left\{1, \tau^4, \kappa^4 \left(\frac{\mu^+ \sigma_*^+}{\mu^- \sigma_*^-}\right)^2, \kappa^2 \left(\frac{\mu^+}{\mu^-}\right)^2 \left(\frac{\sigma_*^+}{\sigma_*^-}\right)\right\} \cdot \max\left\{\frac{w^4(\mathcal{C})}{m}, \frac{m^3}{w^2(\mathcal{C})}, m^2\right\}$, and a feasible initialization $\hat{\boldsymbol{\theta}}_{(0)}$ satisfies $\|\hat{\boldsymbol{\theta}}_{(0)} - \boldsymbol{\theta}_*\|_2 \leq \sqrt{\frac{\sigma_*^-}{\mu^+}}$, then with probability at least $1 - C_2 \exp\left(-C_1 \cdot \min\left\{w^2(\mathcal{C}), m\right\}\right)$, the error bound (30) holds for $\hat{\boldsymbol{\theta}}_{(T)}$ returned by Algorithm 1 with $\rho_n$ and $e_{\min}$ satisfying*

$$\rho_n \leq \frac{C_3 \kappa \mu^+}{\mu^- \sqrt{\sigma_*^- \operatorname{Tr}(\boldsymbol{\Sigma}_*^{-1})}} \cdot \frac{w(\mathcal{S})}{\sqrt{n}} \leq \frac{1}{2} , \qquad e_{\min} \leq \frac{C_4 \kappa \sqrt{\mu^+}}{\mu^- \sqrt{\operatorname{Tr}(\boldsymbol{\Sigma}_*^{-1})}} \cdot \frac{w(\mathcal{S})}{\sqrt{n}} \cdot \frac{1 + \delta_n}{1 - \rho_n} , \tag{S.30}$$

*where $\delta_n$ is the same as the one given in Theorem 1.*

*Proof:* The improved error bound directly follows from the deterministic bound in Theorem S.3 and the probabilistic bounds for $\alpha_n^-, \alpha_n^+$ (Lemma S.4), $\delta_n$ (Proposition S.2), $\gamma_n$ (Lemma S.5) and $\beta_n(\mathcal{M}(e_0))$ with $e_0 = \sqrt{\frac{\sigma_*^-}{\mu^+}}$ (Lemma S.7). ∎