[Reviews · NeurIPS 2018]

Reviewer 1



This paper studies the multi-response regression model, and in particular, the alternating minimization algorithm for the problem . In contrast to prior work, this paper makes the following improvements: 1. It does not require the resampling assumptions that usually abound in showing results on alternating procedures. With good initialization, the procedure without resampling is able to achieve what is usually the minimax rate of such problems. 2. It shows that an arbitrarily initialized problem will still exhibit convergence, but with a worse rate. 3. It allows for the unknown parameter to lie in a possibly non-convex set, and for the noise to be sub-Gaussian. The major technique is that of generic chaining, which allows the authors to prove bounds that hold uniformly over all iterates. I liked the paper and the result overall seems quite interesting, in particular points 1 and 2 above. It seems as though the techniques could also translate to other problems. While I did not read the proofs very carefully, they seem quite reasonable overall given the deterministic conditions that the authors posit. On the negative side, the fact that proof for Alternating Minimization are known to exist without re-sampling means that the paper would benefit from a comparison of the current techniques to those used in the past; if the techniques themselves are not so different, then analyzing AltMin in this particular setting without resampling is not particularly impactful or insightful. Concerns: 1. The authors could de-emphasize point 3 above. It does not particularly impress me, since the authors require an exact solution to each iteration of the parameter estimation problem anyway, irrespective of non-convexity, and this can only be provided in some settings. Similarly, most Gaussian upper bounds can be translated to sub-Gaussian upper bounds since the required behavior of the stochastic process is similar in the two cases. 2. It would be good to compare these results to other results on alternating minimization that do not require resampling (e.g., see the references below). Minor: Line 131: The function could be misinterpreted. Consider stating as d_1 (respectively d_2) to signify two clauses. [1] I. Waldspurger, “Phase retrieval with random Gaussian sensing vectors by alternating projections” IEEE Trans. on Inf Theory, 2018. [2] M. Soltanolkotabi. Algorithms and theory for clustering and nonconvex quadratic programming. PhD thesis, Stanford University, 2014. Post-rebuttal: I have read the response, and maintain my score. This is a strong paper with interesting results.

Reviewer 2



The authors present an improved analysis for the alternative minimization for structured multi-response regression problem. They provide a new way of analyzing the problem and improve on prior work in the domain. The work without the resampling assumption and their analysis work for general sub-Gaussian noise. Moreover, they allow the structuring function to be non-convex and show that the AltMin with arbitrary initialization can achieve the same level of error as the well-initialized one. Although I'm not an expert in the field, I feel this paper is well written and pushes the boundaries of what is known. Moreover, their analysis pertains to what practitioners actually do in real situations. I do have some concerns and hence I would vote for a marginal accept. My details comments are given below. 1. In the introduction, the authors mention real-world applications but only name a few very wide fields. It would be better if they can give some concrete examples and follow up in the experimental section on those datasets. 2. This might be a general comment, since I am unfamiliar with the field. Do the authors have any idea about how the AltMin procedure maps to the EM algorithm that has proven statistical consistency? If so, it would be good to add in a remark so that it appeals to a wider statistical audience. 3. Line 162-163: The authors have assumed that \hat{\theta} can be solved globally despite the potential non-convexity of f. A little more light on such a statement is warranted and would be helpful especially since the paper they cite does not clearly claim as such. 4. The choice of the distance function on \Sigma seems a bit adhoc. Can the authors explain a bit of the reasoning behind such a choice? 5. Since I'm not too familiar with the domain, the assumption on line 180-181 seems a bit weird. Since \Gamma_v depends on X, how do we guarantee that minimum and the maximum eigenvalues are bounded away from 0 and \infty respectively? Minor Comments: 1. Typo in Line 112. Please remove "the" before "statistical models" -> under the assumed statistical models. 2. Please use a different notation for a the \psi_2 norm. The ||| \cdot |||_{\psi}_2 is defined for real numbers. For forcing vectors and matrices with the same notation is a bit confusing. Please change the notation on the LHS of eq(14) and (15) so that the definitions remain consistent. 3. Typo in Line 211: Please remove the repeated use of "significant"

Reviewer 3



This paper proposes a new approach to analyzing alternating minimization algorithms. The main idea is to obtain uniform control the error of one set of iterates in terms of the error of the complementary set of iterates. The benefit of uniform control is the analysis applies to non-resampled alternating minimization algorithms. It's hard to make recommendations beyond presentation of the results without reading the proofs carefully, but the results are certainly first rate. Regarding the presentation of the results, it would be ideal if the authors can state a version of Lemma 2 (or refer to the analogous result in the appendix) in the discussion at the end of Section 3.2. Is the improvement in Theorem 2 from Theorem 1 a result of an improved Lemma 2 if we only restrict to well-initialized $\Sigma(\theta)$'s?